

# Understanding the customer experience in human-computer interaction: a systematic literature review

Daniela Quiñones and Luis Rojas

Escuela de Ingeniería Informática, Pontificia Universidad Católica de Valparaíso, Valparaíso, Valparaíso, Chile

## ABSTRACT

**Background:** In recent years, there has been an increasing interest in customer experience (CX) and its relation to the human-computer interaction (HCI) field. The CX is different depending on the domain in which it is studied, and therefore its dimensions may vary.

**Methodology:** This research presents an extensive review of 122 studies related to CX definitions and dimensions that have been proposed in different domains, including an analysis from an HCI perspective. The guidelines proposed by *Kitchenham & Charters (2007)* were used, complementing the review with a snowballing approach.

**Results:** We identified 71 CX definitions (where 14 definitions highlight HCI aspects), 81-dimensional proposals (where 24 proposals contain HCI aspects), and 39 application domains (where 18 domains cover topics related to HCI). However, we did not find CX definitions or dimensions directly focused on HCI. Based on the results, a novel CX definition and dimensions—focused on the HCI area—are proposed and activities that the authors should perform when proposing new CX dimensions in domains related to HCI are suggested, *i.e.*, domains that involve the interaction of a user (or customer) with a software product.

**Conclusions:** Implications for future practice focus on facilitating the understanding of the CX concept and its relationship with HCI; recognizing the key CX dimensions for different domains and how they relate to HCI dimensions; and helping in the creation of new CX dimensions by suggesting activities that can be performed. The results show that there are opportunities for HCI/CX researchers and practitioners to propose new dimensions of CX for a domain related to HCI, develop instruments that allow the evaluation of CX from an HCI point of view, and perform reviews on a particular domain relevant to HCI but less studied.

## INTRODUCTION

In recent years, there has been increasing interest in customer experience (CX) and its relation between user experience (UX) and human-computer interaction (HCI). The experience at the interaction level is analyzed from a UX approach and addresses designing the experience of a single interaction that a user has with a company to perform a task (*Salazar, 2019a*). In each of these interactions, the user has a specific experience that is only

Corresponding author
Daniela Quiñones,
daniela.quinones@pucv.cl

a small part of the relationship between the customer and the company. In this sense, it is relevant to study the next level of experience: the journey and customer experience (CX). Most journeys consist of a series of interactions with various products, systems, and/or services offered by a company. Providing a good CX presents unique design challenges, so it is relevant to know what this concept means, what its dimensions are, and how they vary in different domains.

CX is generally defined as customers' internal and subjective response to any direct or indirect contact with the company (*Meyer & Schwager, 2007*). Companies are more aware of considering the user and customer when creating new software products. Users and customers use different products, systems, or services offered by a brand, and every moment they interact with them, they have an experience (positive or negative). This experience influences their intent for future use and recognizing the CX dimensions helps to design better software products, and more effectively evaluate the existing ones.

Several authors have proposed CX definitions and dimensions for different domains, as each one has particular characteristics of the specific context. However, most studies only focus on applying the concept of CX in a general way without using a truly appropriate CX definition or dimensions for their studies. In this sense, it is important to know what is meant by CX, what CX dimensions define it, and how these vary according to the application domain, especially in the HCI field. In addition, there are no studies that analyze the CX from an HCI perspective; specifically, there is no reference to CX definitions, CX dimensions, or domains that directly cover aspects of the HCI field. To improve both the CX and the design of better system products, it is necessary to increase knowledge by understanding what the CX is and the dimensions that characterize a certain domain, particularly in HCI. Considering these dimensions, both researchers and practitioners can detect what elements to introduce into their business models and innovate to create more valuable software products.

This research presents a systematic literature review of the concept of CX to identify the multiple definitions for specific domains that have been proposed in the literature. In addition, it attempts to recognize the different attributes, factors, dimensions, or components used to contextualize the CX and how this applies in different contexts and domains. We analyze and discuss the most popular CX definitions and dimension proposals, research domains, and publication years, including an analysis from the HCI perspective. To guide the systematic review, we proposed the following four research questions:

- RQ1: What is the customer experience?
- RQ2: What dimensions define the customer experience?
- RQ3: In which domains are the customer experience concept used?
- RQ4: What is the relation between the customer experience and human-computer interaction?

The research questions were established in such a way that they allowed reviewing the different domains in which CX has been studied, including HCI. Thus, we did not leave out

studies that, although they were not framed in the HCI area (as the main focus), incorporate components related to the design, evaluation, and/or implementation of interactive software products. Based on the results obtained, a novel CX definition and dimensions—focused on HCI—are proposed, and activities that the authors should perform when proposing new CX dimensions in domains related to HCI are suggested, *i.e.*, domains that involve the interaction of a user (or customer) with a software product (interactive computing systems), such as hotels, retail, airlines, and telecommunications.

A new CX definition and dimensions from an HCI perspective are important and useful because: (1) there is no comprehensive CX definition that integrates key elements related, such as touchpoints, channels, dimensions, customers, and products/systems/services. A thorough definition will help HCI/CX researchers and practitioners understand that CX involves a holistic view. It will also help to better understand what elements to study and analyze when researching CX; and (2) there is no definition that directly links HCI to CX. A CX definition that incorporates aspects of HCI will allow emphasizing the interaction of the user (customer) with a software product, knowing which dimensions are involved and their characteristics, facilitating the understanding and evaluation of such interaction. This systematic review reduces the existing gap between HCI and CX. The proposed definition and dimensions of CX from an HCI perspective can be valuable for HCI/CX researchers and practitioners when proposing methods and/or instruments to evaluate CX in specific domains, especially in the HCI area.

The article is structured as follows: "Background" presents the key concepts related to CX definitions and dimension proposals; additionally, it analyzes other literature reviews about CX. "Research Method" introduces the research method used for the systematic literature review. "Study Selection" reports the selected studies. "Results" summarizes the findings of the systematic review. "Discussion" analyzes the results and deepens the relation between CX and HCI. Finally, "Conclusions" shows the conclusions.

## BACKGROUND

The definitions of CX, its dimensions, and how it is understood in different domains are presented below. In addition, the relation between HCI, CX, UX, and the related work is described.

### Customer experience

Recently, the concept of CX has received the interest of practitioners and researchers from several domains and involves studying customers to achieve a competitive advantage in several industries. Throughout the years, numerous definitions have been proposed to define CX in both general and specific domains. For instance, *Meyer & Schwager (2007)* define CX as "the internal and subjective response customers have to any direct or indirect contact with a company".

Similarly, *Verhoef et al. (2009)* state that "the customer experience construct is holistic in nature and involves the customer's cognitive, affective, emotional, social and physical responses to the retailer. This experience is created, not only by those elements that the retailer can control (*e.g.*, service interface, retail atmosphere, assortment, price) but also by

elements that are outside of the retailer's control (*e.g.*, the influence of others, purpose of shopping)".

For *Jain, Aagja & Bagdare (2017)*, CX means "the aggregate of feelings, perceptions, and attitudes formed during the entire process of decision-making and consumption chain involving an integrated series of interaction with people, objects, processes, and environment, leading to cognitive, emotional, sensorial and behavioral responses". The concept of CX is used by *Belabbes & Oubrich (2018)* to refer to "the consequence of the physical and the emotional contact that the customer has with the company, its offers or surroundings through the customer journey. It results in feelings, attitudes, and behaviors that the customer expresses in the form of satisfaction, loyalty, recommendation, and purchase".

The definitions presented above highlight different approaches to the concept. *Meyer & Schwager (2007)* stress the internal and subjective components that arise from both direct and indirect contact with a company. *Verhoef et al. (2009)* affirm that CX is a holistic concept that encompasses the total experience throughout the customer's purchase journey and emphasizes the existence of elements outside the control of a company. *Jain, Aagja & Bagdare (2017)* indicate that customer responses are produced by a series of interactions with different factors (people, objects, processes, and the environment). *Belabbes & Oubrich (2018)* mention that CX generates distinct reactions in customers that result in certain consequences and outcomes (satisfaction, loyalty, recommendation, and purchase). As this systematic review aims to analyze and discuss different CX definitions, a deeper comparative analysis of the different definitions proposed over the years is presented in the Discussion section.

Multiple dimensions, attributes, and/or components have been proposed to understand and characterize CX over the years. These dimensions emerged in the marketing domain, where *Schmitt (1999)* was one of the first researchers to emphasize the relevance of CX, introducing five dimensions: (1) sense, related to the senses to create sensory experiences; (2) feel, associated with feelings and emotions to create affective experiences; (3) think, linked to the intellect to create cognitive and problem-solving experiences; (4) act, involves customer behaviors and lifestyles by targeting their physical experiences; and (5) relate, refers to individual's personal and private feelings in a broader social system.

Furthermore, several studies propose dimensions based on Schmitt's proposal, such as *Gentile, Spiller & Noci (2007)*, which propose six dimensions for CX: (1) emotional, associated to the emotions and feelings resulting from an interaction with a company and its products; (2) sensorial, related to the sensory experience through the five senses; (3) pragmatic, related to the concept of usability and the practical act of doing something; (4) cognitive, linked to conscious mental processes and creativity or problem-solving abilities; (5) relational, associated with the social context, relationships, or the ideal of self; and (6) lifestyle, linked to values, beliefs, the adoption of a lifestyle and behavior.

The dimensions proposed by *Gentile, Spiller & Noci (2007)* are defined in a general way, allowing their application to different domains. Nevertheless, some authors propose dimensions for CX in specific domains. For instance, *Nambisan & Watt (2011)* suggest four dimensions for CX in online communities: (1) pragmatic; (2) hedonic; (3) sociability;

and (4) usability. In another domain, *Shin, Cho & Lee (2019)* present four dimensions for CX in banks: (1) usefulness; (2) convenience; (3) employee-customer engagement; and (4) security. In "Proposals for Customer Experience Dimensions (RQ2)" and "Discussion" an analysis and discussion of different CX dimensions found in this systematic review are presented.

## Relationship between human-computer interaction, customer experience, and user experience

Human-computer interaction (HCI) is a "discipline concerned with the design, evaluation, and implementation of interactive computing systems for human use and with the study of major phenomena surrounding them" (*Hewett et al., 1992*). It involves users (customers) and their interaction with a software product, therefore, "the HCI interest in CX becomes obvious" (*Rusu et al., 2018*). HCI is considered an interdisciplinary area and is composed of five interrelated aspects or dimensions (*Hewett et al., 1992*):

1. Nature of human-computer interaction: related to the overview and theoretical framework about topics in HCI.
2. Use and context of the system: related to the software uses and the context where is used. It includes the social organization (related to the human as an interacting social being), work, and business context.
3. Human characteristics: related to human information-processing characteristics (cognitive models), human communication and interaction, and ergonomics.
4. Computer system and interface architecture: related to the technical construction of devices, computer graphics (design, color), and user interface.
5. Software development process: related to the construction of the human interface, design process, software implementation, and evaluation techniques.

Some studies consider CX and UX synonymous (*Boureanu, 2017*). They share aspects and practices for achieving customer satisfaction and customer loyalty. However, both disciplines have different approaches and degrees of interaction. As stated by Salazar, "the term customer experience is used to describe the broadest scope of the user experience" (*Salazar, 2019b*). UX focuses on facilitating the use of products, systems, or services to users through intuitive tasks, or quick and easy ways to obtain information. This is achieved by the joint effort of practitioners related to design (UX designer), communications (UX writer), research (UX researcher), and technology (UX engineer) tasks. On the other hand, CX focuses on achieving a pleasant, professional, and valuable interaction of customers with a company and its representatives. This is accomplished by concentrating the efforts of different practitioners (marketing, operations, strategy, and business experts) on customer service, advertising, the buying process, product delivery, and the UX of each product, system, or service. Therefore, UX refers to a user's perceptions of a single product, system, or service, while CX includes different interactions with a company through multiple products or systems in the customer journey (*Rusu et al., 2020*; *Lewis, 2014*).

UX is contained within CX, and it is generally accepted that UX is an extension of the usability concept. This can be observed in the "usable" attribute proposed by *Morville (2004)*. On the other hand, Nielsen mentions that "web design and usability are subsets of the greater discipline of human-computer interaction (HCI)" (*Nielsen, 2002*). The HCI field lies at the intersection between the social and behavioral sciences and computer and information technology (*Carroll, 2003*). As *Preece et al. (1994)* stated in their study, the HCI discipline is broader than just the interface design, and it considers all aspects related to the interaction. Consequently, as a result of these implications, it can be stated that CX is one of the multiple disciplines within the multidisciplinary field of HCI.

It is important to study UX and HCI from a CX perspective. For that reason, this systematic literature review focuses on CX, since UX and HCI are usually not considered holistically in many organizations, leaving aside interactions with different channels, devices, and technologies that may be related to software systems (*Salazar, 2016*).

## Related work

Diverse literature reviews covering different topics within CX research were found, including: (1) studies that introduce findings or theoretical perspectives, (2) studies that differentiate the concepts of CX and service experience, and (3) studies that summarize definitions and dimensions in a specific domain. However, no reviews were found that analyzed the CX from an HCI perspective.

*Lipkin (2016)* carried out research identifying three theoretical perspectives (stimulus-based, interaction-based, and sense-making-based) and three contextual lenses (dyadic, service ecosystem, and customer ecosystem) that explain CX formation in service settings. *Kranzbühler et al. (2018)* categorize CX research on two levels (static and dynamic CX) and investigate both levels from two theoretical perspectives (organizational and customer).

Similarly, *Adhikari & Bhattacharya (2016)* introduce two different but interrelated streams related to CX: (1) experience as a product attribute or a complete product, and (2) CX created due to customer interaction with the physical environment or people. In addition, they propose a framework related to three main aspects of CX (antecedents, creation, consumption of experience, and effect of CX). *Becker & Jaakkola (2020)* recognize eight literature fields that study CX in marketing, grouped into two research traditions: (1) CX as a response to managerial stimuli, and (2) CX as a response to consumption processes.

Last, *De Keyser et al. (2020)* introduce a formal nomenclature for the CX identifying several components comprised of three building blocks (touchpoints, context, and qualities). *Silva et al. (2020)* perform a literature analysis using bibliometric techniques to provide a map of CX research revealing three clusters of research (service quality, service encounter, and service-dominant logic), which constitute the foundations of the CX research field.

Concerning studies focused on differentiating the concepts of CX and service experience, *Jain, Aagja & Bagdare (2017)* review the literature on CX regarding the emergence, development, and theorization of the concept, explaining the similarities,

differences, and relationship between service experience and CX. Later, *Bueno et al. (2019)* perform a systematic review of CX in service discussing both concepts and distinguishing how they have been measured in relevant publications in the marketing field. Likewise, *Cano, Rusu & Quiñones (2020)* conducted a similar study focused on identifying methods to evaluate CX and behavior.

The main differences between the studies mentioned above and this study are: (1) the distinct focus because these revisions are focused on proposing different theoretical perspectives, differentiating concepts presented in the literature, or identifying evaluation methods rather than identifying CX definitions and dimensions over time and (2) narrow scope, as the findings identified apply only to certain specific domains. The CX concept and dimensions from the point of view of HCI were analyzed and discussed.

Respecting studies focused on summarizing CX definitions and dimensions in a specific domain, *Mahr, Stead & Odekerken-Schröder (2019)* review the concepts and theories underlying CX (named by the authors customer service experience) and its five dimensions (physical, social, cognitive, affective, and sensorial) across the service and marketing domains using a text mining approach. *Waqas, Hamzah & Salleh (2020)* categorized and summarized CX research, identifying different conceptualizations, dimensions, antecedents, consequences, and theoretical perspectives of CX in marketing. *Godovykh & Tasci (2020)* analyze empirical and conceptual literature on experience, proposing an experience model composed of four components (emotional, cognitive, sensorial, and conative), and suggesting methods/techniques to measure CX in tourism at the pre-visit, on-site, and post-visit stages. However, this article does not clearly differentiate the concepts of experience and CX by mixing both definitions and dimensions. Table 1 shows a summary of the main characteristics and shortcomings of the related works reviewed.

Surprisingly, although research questions are the most important activity in a systematic review (*Kitchenham & Charters, 2007*), multiple studies reviewed (see Table 1) do not define them, making it difficult to guide, understand, and delimit the study. In addition, although the related studies mention the keywords used in their search strategy, none of these studies show the search strings used to perform the systematic review, so it could be difficult to find the reviewed studies and guarantee the replicability of the research. Finally, several studies did not clearly mention the inclusion and/or exclusion criteria used to select the reviewed studies, which could make it difficult to identify the studies that answer the research questions, ensuring objectivity and transparency and reducing the likelihood of bias (*Kitchenham & Charters, 2007*).

The terms "human-computer interaction" and "HCI" were not presented or discussed in any of the literature reviews focused on CX. Therefore, there is no reference to CX definitions, CX dimensions, or domains that directly cover aspects of the HCI field. Based on the above, a systematic literature review to complement CX research in HCI was conducted by (1) linking the concept of CX with HCI; (2) proposing a CX definition and dimensions from an HCI perspective; (3) associating HCI aspects with CX dimensions; and (4) providing a series of steps to propose new dimensions of CX for a domain related to HCI. To achieve this, it is necessary to identify (1) what is meant by CX; (2) what components or dimensions define CX; and (3) in which domains CX is used or considered.

**Table 1 Characteristics and shortcomings of the related works reviewed.**

| Authors | Scope | Years covered | Databases considered | Are research questions included? | Are search strings explicitly documented? | Are inclusion and/or exclusion criteria clearly explained? |
|---|---|---|---|---|---|---|
| Lipkin (2016) | Focused on CX in services | 1998 to 2015 | EBSCO, Emerald, and ABI/Proquest | Yes | No | Yes |
| Kranzbühler et al. (2018) | Focused on CX | 1982 to 2016 | EBSCO's Business Source Elite | No | No | Yes |
| Adhikari & Bhattacharya (2016) | Focused on CX in tourism sector | 1970 to 2014 | EBSCO, JSTOR, and ABI ProQuest | Yes | No | No |
| Becker & Jaakkola (2020) | Focused on CX in marketing | Not mentioned | EBSCO and Science Direct | Yes | No | Yes |
| De Keyser et al. (2020) | Focused on CX in business and management | 1982 to 2020 | Social Science Citation Index | No | No | Yes |
| Silva et al. (2020) | Focused on CX | 1991 to 2018 | Social Science Citation Index | No | No | No |
| Jain, Aagja & Bagdare (2017) | Focused on CX and Service experience | 1990 to 2015 | Emerald, EBSCO, and Science Direct | Yes | No | No |
| Bueno et al. (2019) | Focused on CX in the service sector and marketing field | Not mentioned | Web of Science and Scopus | Yes | No | Yes |
| Cano, Rusu & Quiñones (2020) | Focused on CX | 2016–2020 | IEEE Xplore, ScienceDirect, Sage Journals, ACMDL, and Scopus | Yes | No | No |
| Mahr, Stead & Odekerken-Schröder (2019) | Focused on CX in services and marketing | 1994 to 2018 | Thomson Reuters' InCite Citation Index and Web of Science | Yes | No | Yes |
| Waqas, Hamzah & Salleh (2020) | Focused on CX in marketing | 1998 to 2019 | Emerald, Scopus, EBSCO, Springer, and JSTOR | No | No | Yes |
| Godovykh & Tasci (2020) | Focused on CX in tourism sector | Not mentioned | Not mentioned | No | No | No |

## RESEARCH METHOD

A systematic literature review using the protocol proposed by *Kitchenham & Charters (2007)* was carried out between October 2020 and July 2021. The review procedure consisted of three steps with several activities:

1. Planning the review: establish the importance of a systematic literature review and define the research questions and the review protocol (*i.e.*, data sources, search strategy and terms, study selection criteria, study quality, and data extraction and synthesis).
2. Conducting the review: select the articles to review, answer the research questions, introduce the results, present the discussions, and summarize the conclusions.
3. Reporting the review: structure and write up the results of the systematic literature review.

The review was complemented using a snowballing approach. Snowballing refers to using the reference list of a article or the citations to the article to identify additional articles (*Wohlin, 2014*). The approach proposed by *Wohlin (2014)* was utilized, with backward snowballing only to articles that met the inclusion criteria selected. If any article cited in a selected article contains (or probably contains) information that answers the research questions, they were also reviewed.

To choose a article from the examined article, the title, the publication venue, and the authors from the reference list were examined. Articles were not excluded if an author is not known for publishing in the CX or HCI area, but a article may be more likely to be included if the author regularly publishes in the CX or HCI area (*Wohlin, 2014*). A list of all the articles to review was made, and after finishing the examination of the article, the articles obtained by snowballing were reviewed. If any of these articles met our inclusion criteria after reviewing the abstract and keywords, they were selected.

## Data sources

The database sources involved in this review included those considered relevant to scientific research, engineering, computer science, and technology: (1) Science Direct, (2) Scopus, (3) IEEE Xplore Digital Library, (4) ACM Digital Library, and (5) Springer Link. Studies from September 2010 to September 2020 were reviewed.

## Search strategy and terms

The search terms contained "customer" and "experience", as well as the combination of both as "customer experience". We used terms related to the meaning of CX, for example, "definition" and "concept". Finally, we added terms related to the aspects, qualities, or facets of CX, for example, "attributes", "dimensions", "features", "aspects", "components", and "elements". The advanced search form was used to narrow the scope of the search. However, each database has a different syntax and logic in its search strings, *e.g.*, Scopus allows filtering by title, abstract, and keyword typing "TITLE-ABS-KEY" before the search string. ScienceDirect has a field called "Title, abstract, or author-specified keywords". Nevertheless, SpringerLink does not allow filtering results using titles, abstracts, and keywords. The search strings used for each database are shown in Table 2.

## Study selection criteria

Several criteria were defined to determine appropriate articles for inclusion and exclusion from the systematic review. Studies that fulfilled the following criteria were considered:

1. Research studies.
2. Studies that mention CX definitions, as a concept or term.
3. Studies that present or propose CX dimensions, as attributes, features, aspects, components, and/or elements.
4. Studies published between September 2010 and September 2020.

**Table 2 Search strings for each database.**

| Database | Search strings |
|---|---|
| ScienceDirect | ("customer experience") AND (concept OR definition)<br>Title, abstract, keywords: "customer experience" AND Year: 2010–2020 |
| | ("customer experience") AND (attributes OR dimensions OR features OR aspects OR components OR elements)<br>Title, abstract, keywords: "customer experience" AND Year: 2010–2020 |
| Scopus | (TITLE-ABS-KEY (("customer experience") AND (define OR definition)) AND KEY ("customer experience")) AND PUBYEAR > 2009 AND PUBYEAR < 2021 |
| | (TITLE-ABS-KEY (("customer experience") AND (dimensions)) AND KEY ("customer experience")) AND PUBYEAR > 2009 AND PUBYEAR < 2021 |
| IEEE Xplore | ((("All Metadata": customer experience) AND "All Metadata": definition OR concept) AND "Author Keywords": customer experience)<br>Year: 2010–2020 |
| | ((("All Metadata": customer experience) AND "All Metadata": attributes OR dimensions OR features OR aspects OR components OR elements) AND "Author Keywords": customer experience)<br>Year: 2010–2020 |
| ACM Digital Library | [All: "customer experience"] AND [[All: concept] OR [All: definition]] AND [Keywords: "customer experience"] AND [Publication Date: (07/01/2010 TO 07/31/2020)] |
| | [All: "customer experience"] AND [[All: attributes] OR [All: dimensions] OR [All: features] OR [All: aspects] OR [All: components] OR [All: elements]] AND [Keywords: "customer experience"] AND [Publication Date: (07/01/2010 TO 07/31/2020)] |
| SpringerLink | Customer experience AND (definition OR concept)<br>Title: "customer experience" AND Year: 2010–2020 AND Language: English |
| | ("customer experience") AND (attributes OR dimensions OR features OR aspects OR components OR elements)<br>Title: "customer experience" AND Year: 2010–2020 AND Language: English |

5. Studies that present definitions or dimensions of the CX published before 2010, but that are cited as the original source in the reviewed articles or (backward snowballing method).

6. Studies written in Spanish or English.

In addition, the following studies were excluded:

1. Unpublished theses (*e.g.*, undergraduate or master's theses).

2. Studies that include CX in their keywords but do not present definitions or dimensions.

3. Studies not focused on the conceptualization of CX and its related dimensions, such as reports on CX evaluation, design, or case studies.

4. Articles not explicitly focused on the CX, that is, articles that focus on something too specific (*e.g.*, customer retention, antecedent, only one CX dimension) or not directly related to the CX (*e.g.*, only "experience", service experience, hotel factors, experiential marketing).

5. Articles that present inconsistent information within the same document (*e.g.*, (1) in one part the authors present three CX dimensions, and then they present four dimensions; (2) the names of the dimensions change throughout the document).

**Table 3 Criteria for evaluating the quality of studies.**

| No. | Question | Score (yes = 1, partially = 0.5, no = 0) |
|-----|----------|------------------------------------------|
| Q1 | Are the aims of the study clearly stated? | |
| Q2 | Are the scope and context and experimental design of the study clearly defined? | |
| Q3 | Are the variables in the study likely to be valid and reliable? | |
| Q4 | Is the research process documented adequately? | |
| Q5 | Are all the study questions answered? | |
| Q6 | Are the negative findings presented? | |
| Q7 | Are the main findings stated clearly? Regarding creditability, validity, and reliability? | |
| Q8 | Do the conclusions relate to the aim and purpose of the study? Are they reliable? | |

## Quality of the studies

To complement the inclusion and exclusion criteria, criteria were used to assess the quality of the studies to reduce bias and increase internal and external validation (*Kitchenham & Charters, 2007*). To assess the quality, the criteria proposed by *Tummers, Kassahun & Tekinerdogan (2019)* were used, who adapted the criteria from *Kitchenham & Charters (2007)*.

The quality assessment was made by reading the articles completely and applying the criteria presented in Table 3. The quality score assisted in the primary study selection. A three-point scale was applied (yes = 1, partially = 0.5, no = 0) for each criterion found during the selection process. The process described by *Vasiljevic & de Miranda (2020)* was used to analyze each study. The studies were analyzed by the researchers and answered "yes", "partially", or "no" depending on whether the article met the criteria reviewed. After this, each study received a score from 0 to 8. Articles with a score equal to or greater than 4 were included in the final analysis.

For questions Q2, Q4, and Q7 (see Table 3) the emphasis was on evaluating that the articles: (1) describe in detail the CX concept, discussing its definition, or meaning; (2) describe in detail the CX dimensions proposal, including the process used to create the proposal; and (3) present definitions and/or dimensions proposals, with sufficient information to understand them.

## Data extraction and synthesis

For each study selected for the literature review related to CX definitions, information was identified, extracted, and synthesized, including the following seven elements: the authors; the year of the study; the title of the article; the definition of the CX; the main and specific domain in which the concept of the CX is used; the type of the article (conference or journal article); and articles that cite the definition.

For each study selected for the literature review related to CX dimensions, information was identified, extracted, and synthesized, concerning the following eight elements: the authors; the year of the study; the title of the article; the list of dimensions proposed for characterizing the CX, including its definitions and amount; the categories (if applicable) used to group the dimensions proposed; the main and specific domain in which the

dimensions of the CX are used; the way the dimension proposal is presented (as diagram or graphic, table, text, or other); and the reference of articles that cite the dimensions proposal.

The information gathered on the studies was grouped, summarized, and tabulated in two tables based on CX definitions and dimensions. The data were recorded using Excel spreadsheets. The information obtained is analyzed in Section 5 (Results) and Section 6 (Discussion).

## STUDY SELECTION

Based on the established inclusion and quality assessment criteria, 122 studies were recognized as pertinent to the review. Of these articles, 73 presented 71 definitions of CX, while 79 presented 81 dimension proposals to characterize CX. Additionally, 27 studies propose in the same article both a definition and dimensions of CX.

Of the 122 studies, 84 were identified through the search strings defined in the systematic review (see Table 4), while 38 were identified through snowballing. Figure 1 shows a PRISMA (*Page et al., 2021*) flow diagram detailing the study selection process. Table 4 presents a summary of the studies reviewed. A total of 502 articles were excluded after scanning the full text and applying the quality assessment criteria. Several articles included the "customer experience" concept in their keywords; however, they did not address any topic related to CX.

## RESULTS

This section presents the results obtained in the systematic literature review. Each research question is answered below.

### Customer experience definitions (RQ1)

According to the review of the studies, 71 different definitions were proposed for explaining and describing CX. Of these, 46 definitions (64.8%) were detected using the defined queries, whereas 25 of them were detected through snowballing (35.2%, see Fig. 2A). On the other hand, most of the proposed CX definitions have been published in scientific journals (47 studies, 66.2%), and fewer have been published at conferences and books (see Fig. 2B).

Based on the articles reviewed, the first CX definition was proposed in 1999 by *Schmitt (1999)* in the context of experiential marketing. *Schmitt (1999)* states that a customer's experience occurs as a result of interaction or encounter with things, and this experience is affected by sensory and emotional elements.

As shown in Fig. 3, the years with the largest number of publications correspond to 2018 (10 studies), 2015 (nine studies), and 2017 (eight studies). It is possible to observe that over the years, interest in investigating the CX has increased, specifically since 2015. Researchers are gradually including CX in their studies as a relevant factor when improving products or services offered by various companies or organizations.

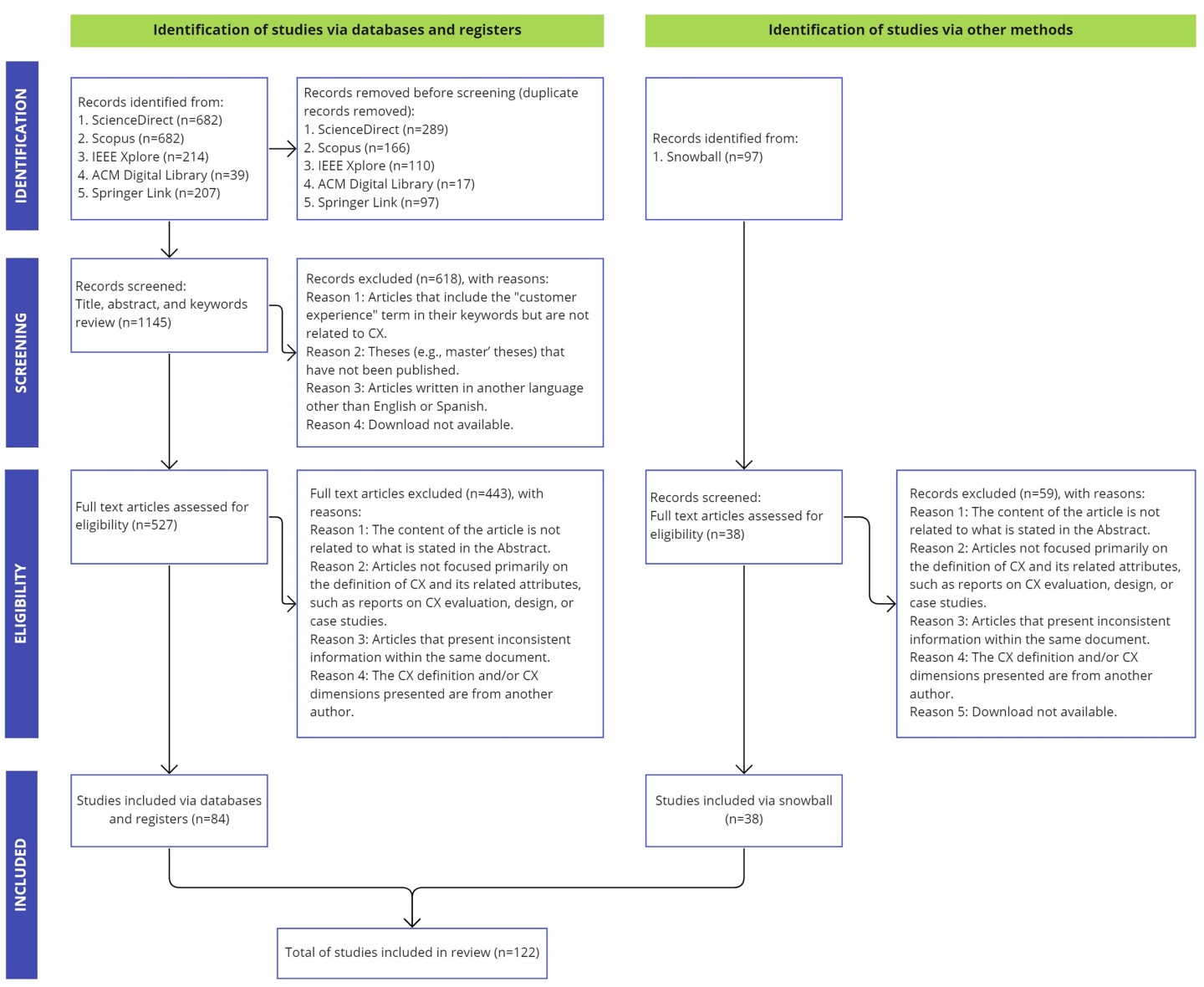

**Figure 1 PRISMA flow diagram with the results of the article selection process.**

**Table 4 Papers selected using the inclusion criteria.**

| Type of study | Number of studies | Description |
|---|---|---|
| Studies related to customer experience definition | 73 | 71 definitions |
| Studies related to customer experience dimensions | 79 | 81 dimension proposals |

Although 71 definitions were identified, some of them highlight the number of citations or times that they have been used as a basis to propose a new definition or to particularize it for a specific domain. Table 5 shows the most commonly used CX definitions.

**A**

**B**

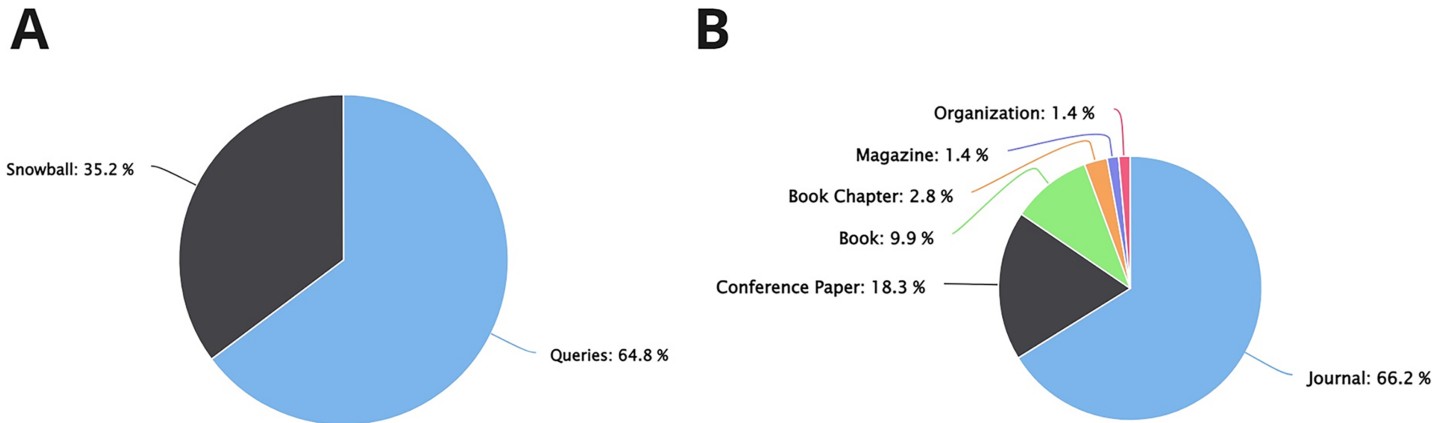

**Figure 2 Information about the studies found related to CX definitions.** (A) CX definitions found by queries *vs* snowball. (B) Type of studies that propose CX definitions.

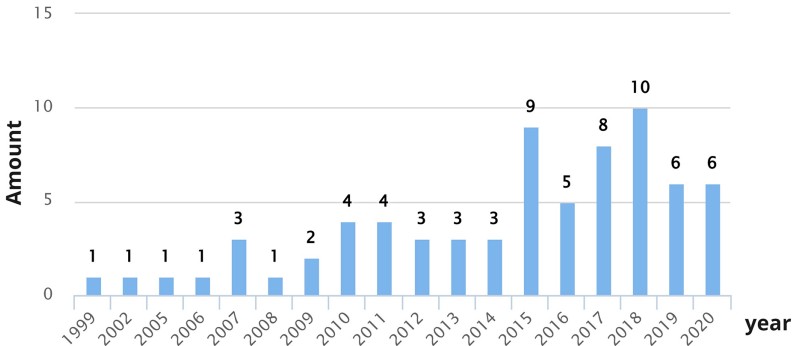

**Figure 3 Number of CX definition proposals per year.**

When reviewing the 71 definitions established to describe CX, it is possible to recognize certain common words, concepts, or terms that are repeated between the different proposals. Figure 4 shows the most commonly used words among the different definitions to describe what CX is. Both the words "customer" and "experience" are used in all definitions, and many times, more than once in the same definition (167 times for the word "customer" and 116 times for the word "experience"). In addition, some authors use the concept of "consumer" in their definition to refer to the user who consumes products or services at different moments of interaction with a brand or company.

As common elements (see Fig. 4), the definitions of CX highlight the "interaction" of the customer with the "products" or "services" offered by a brand or "company"; interaction in which "emotional", "cognitive", and "social" components are involved. The customer "perceives" the product or service in a specific way, causing them to have a "response" to it. The interaction should generate "value" for the customer, which has a positive or negative impact on their experience. The above is very relevant from an HCI perspective, where the interaction of the user, in this case, the customer, with a system or software is essential.

**Table 5 CX definitions most used.**

| Definition | Authors and year | Amount of cites |
|---|---|---|
| "CX is the internal and subjective response customers have to any direct or indirect contact with a company. Direct contact generally occurs in the course of purchase, use and service and is usually initiated by the customer. Indirect contact most often involves unplanned encounters with representations of a company's products, services, or brands and takes the form of word-of-mouth recommendations or criticisms, advertising, new reports, reviews, and so forth". | *Meyer & Schwager (2007)* | 21 |
| "The CX construct is holistic in nature and involves the customer's cognitive, affective, emotional, social and physical responses to the retailer. This experience is created not only by those elements which the retailer can control, but also, by elements that are outside of the retailer's control. CX encompasses the total experience, including the search, purchase, consumption, and after-sale phases of the experience, and may involve multiple retail channels". | *Verhoef et al. (2009)* | 12 |
| "The CX concept is defined as an evolution of the concept of relationship between the company and the customer. The CX originates from a set of interactions between a customer and a product, a company, or part of its organization, which provoke a reaction. This experience is strictly personal and implies the customer's involvement at different levels (rational, emotional, sensorial physical and spiritual)". | *Gentile, Spiller & Noci (2007)* | 11 |
| "Experiential marketing focuses on CX. Experiences occur as a result of encountering, undergoing, or living through things. Experiences provide sensory, emotional, cognitive, behavioral, and relational values that replace functional values". | *Schmitt (1999)* | 10 |
| "The CX is the customer's subjective response to the holistic direct and indirect encounter with the firm, including but not necessarily limited to the communication encounter, the service encounter, and the consumption encounter". | *Lemke, Clark & Wilson (2011)* | 4 |
| "CX is a multidimensional construct focusing on a customer's cognitive, emotional, behavioral, sensorial, and social responses to a firm's offerings during the customer's entire purchase journey". | *Lemon & Verhoef (2016)* | 4 |

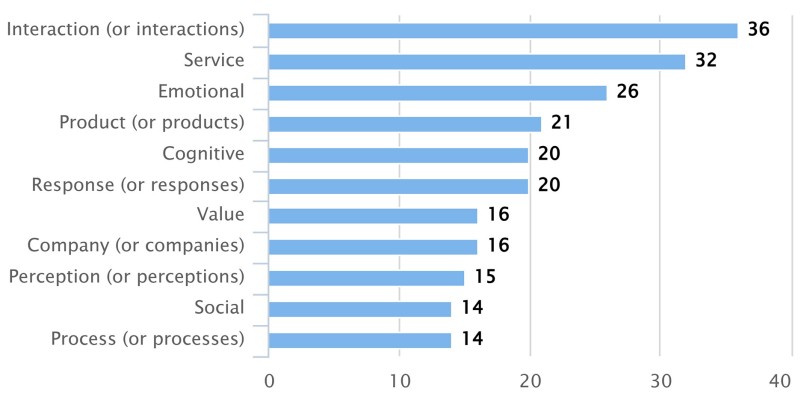

**Figure 4 Most used words among the different definitions to describe the CX.**

## Proposals for customer experience dimensions (RQ2)

Approximately 79 studies that presented 81 proposals of CX dimensions in several domains were found. Among these proposals, most of these—specifically 55 proposals (69.6%)—were obtained through the defined queries for the different databases (see Fig. 5A). These proposals come from studies of different types. It was identified that 59 of 79 studies found (73.4%) come from scientific journals (see Fig. 5B). It was observed that

**A**                                          **B**

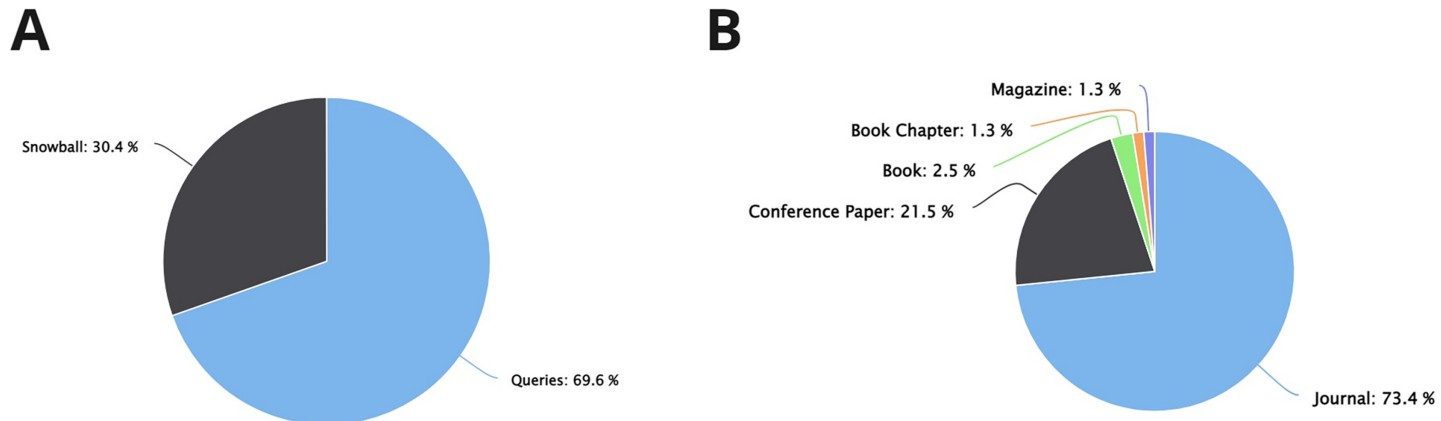

**Figure 5  Information about the studies found to be related to CX dimensions.** (A) CX dimensions found by queries *vs* snowball. (B) Type of studies that propose CX dimensions.

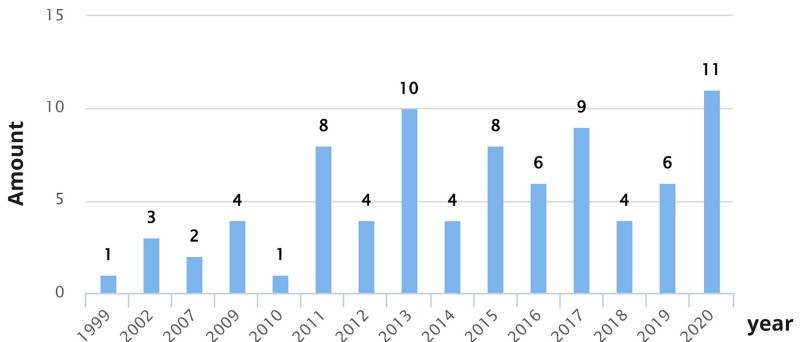

**Figure 6  Number of CX dimension proposals per year.**

authors have a preference for proposing CX dimensions in journals rather than conferences or others.

These studies have been published over the years and different proposals were found between 1999 and 2020. As shown in Fig. 6, the years with the largest number of publications correspond to 2020 (11 studies), 2013 (10 studies), and 2017 (nine studies). The first formal proposal for CX dimensions (named by the author modules) came from *Schmitt (1999)* in 1999 in an experiential marketing context. Publications related to proposals of CX dimensions gradually increased from 2011. Although there are years in which fewer dimension proposals were published (2012, 2014, 2018), an increase in studies can be noted as of 2011.

Within these studies, there is no clear differentiation by the authors to define these dimensions, since terms such as components, attributes, or factors are used often interchangeably or as synonyms. However, when performing this systematic literature review, the most relevant articles (51 studies, 63%) were identified mainly using the name "dimensions" (see Fig. 7).

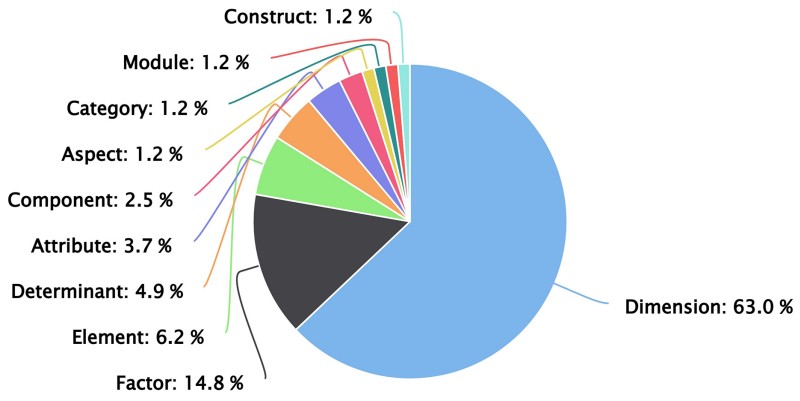

**Figure 7  Common name used to define proposals.**

**Table 6  CX dimensions most used.**

| Dimensions | Authors and year | Amount of cites |
|---|---|---|
| Sense, Feel, Think, Act, Relate | *Schmitt (1999)* | 18 |
| Sensorial, Emotional, Cognitive, Pragmatic, Lifestyle, Relational | *Gentile, Spiller & Noci (2007)* | 12 |
| Social environment, Service interface, Retail atmosphere, Assortment, Price, Promotions | *Verhoef et al. (2009)* | 9 |
| Sensory, Affective, Intellectual, Behavioral | *Brakus, Schmitt & Zarantonello (2009)* | 6 |

Among the several studies that propose dimensions for CX, different proposals are widely recognized by other authors (see Table 6). The dimensions proposed by *Schmitt (1999)* were the most cited among the selected studies, mentioned in 18 of 79 articles, followed by the proposal of *Gentile, Spiller & Noci (2007)* with 12 citations.

Within the selected studies, several CX dimensions were proposed in different domains, with dimensions that are used more frequently than others. The most mentioned dimension of the different studies was "social" and its variations, such as "social" experience or "social" interaction, which was presented in 14 proposals, followed by "cognitive", and "emotional" (see Table 7). These three dimensions are directly related to the HCI aspects of "use and context of software" and "human characteristics" (*Hewett et al., 1992*).

## Domains in which the customer experience concept is used (RQ3)

Various domains are subdivided into smaller domains (named in this article "specific domains"), focused on a particular area within that domain. 79 specific domains grouped into 39 main domains in which different CX definitions and CX dimensions are proposed. Specifically, CX definitions have been proposed for 24 main domains and 45 specific domains; and CX dimensions have been proposed for 31 main domains and 57 specific domains. Most of these domains cover different aspects of social interactions and are related to the HCI field (see Table 8).

Twenty-three CX definitions have been proposed in a general way, that is, definitions that can be used for any domain, regardless of its characteristics and/or application area

**Table 7 Most mentioned CX dimensions and their variations.**

| Dimension | Amount | Variations |
|---|---|---|
| Social | 14 | Social environment, Social experience, Social interaction, Social presence, Social response, Social value |
| Cognitive | 12 | Cognitive experience, Cognitive experiential State, Cognitive fit, Cognitive response, Cognitive value |
| Emotional | 10 | Emotional experience, Emotional fit, Emotional support, Emotive experience |
| Environment | 10 | Environment experience, Outer environment, Overall environment, Physical environment |
| Sense | 10 | Sensorial, Sensorial experience, Sensorial response |
| Accessibility | 8 | – |
| Hedonic | 8 | Hedonic enjoyment, Hedonic experience, Hedonic service, Hedonism, Online hedonic elements |
| Relate | 8 | Relational, Relational experience, Relationship experience |
| Convenience | 7 | – |
| Price | 7 | Pricing, Pricing better than competitor |
| Affective | 6 | Affective experience, Affective Experiential State, Affective response |
| Interactivity | 6 | Customer interaction, Interactions with other customers, Interactions with staff |
| Perception | 6 | Perceived control, Perceived enjoyment, Perceived experience, Perceived luck, Perceived quality |
| Behavioral | 5 | Behavior experience, Behavioral |
| Moments-of-truth | 5 | – |
| Sensory | 5 | Sensory appeal, Sensory experience |
| Trust | 5 | Trust experience |

**Table 8 Number of CX definitions and dimension proposals by domain.**

| Domain | CX definitions | | CX dimensions | |
|---|---|---|---|---|
| | Number of definitions | References | Number of dimension proposals | References |
| Airline | 1 | *Laming & Mason (2014)* | 1 | *Chauhan & Manhas (2017)* |
| Automobile | 1 | *Zhang et al. (2011)* | – | – |
| Bank[1] | 3 | *Komulainen & Makkonen (2018), Fernandes & Pinto (2019), Klaus & Maklan (2012), Klaus & Maklan (2013)* | 9 | *Shin, Cho & Lee (2019), Fernandes & Pinto (2019), Klaus & Maklan (2012), Klaus & Maklan (2013), Chahal & Dutta (2015), Garg et al. (2012), Sharma & Chaubey (2014), Loureiro & Sarmento (2017), Wasan (2018), Maklan & Klaus (2011), Klaus et al. (2013)* |
| Blog | 2 | *Hsu & Tsou (2011), Chen & Lin (2015)* | 1 | *Hsu & Tsou (2011)* |
| Brand | 1 | *Ghose (2009)* | 2 | *Brakus, Schmitt & Zarantonello (2009), Wang et al. (2017)* |
| Business | 3 | *Lemke, Clark & Wilson (2011), Seppanen & Laukkanen (2016), Behare, Waghulkar & Shah (2018)* | – | – |
| Casino | – | – | 1 | *Wong & Wu (2013)* |
| Construction | – | – | 1 | *Al-Fadly (2020)* |
| Digital Platform | – | – | 1 | *Saberian et al. (2020)* |
| Digital technology | – | – | 1 | *Parise, Guinan & Kafka (2016)* |

| Domain | CX definitions | | CX dimensions | |
| --- | --- | --- | --- | --- |
| | Number of definitions | References | Number of dimension proposals | References |
| E-Commerce[2] | 3 | *Klaus (2013), Pei et al. (2015a, 2015b), Liu et al. (2017)* | 3 | *Klaus (2013), Pei et al. (2015a, 2015b)* |
| Education | 1 | *Dou et al. (2019)* | – | – |
| Food and Wine event | – | – | 1 | *Liu, Sparks & Coghlan (2017)* |
| Fresh Products APPs | – | – | 1 | *Su et al. (2019)* |
| Fuel and Service Station | 1 | *Klaus & Maklan (2013)* | 1 | *Klaus & Maklan (2013)* |
| General | 23 | *Meyer & Schwager (2007), Jain, Aagja & Bagdare (2017), Gentile, Spiller & Noci (2007), (Lipkin (2016), Becker & Jaakkola (2020), De Keyser et al. (2020), Waqas, Hamzah & Salleh (2020), Lee, Ka-hyun Lee & Choi (2018), Shaw & Ivens (2002), De Keyser et al. (2015), Boureanu (2017), Klaus (2015), Bolton et al. (2018), Shaw & Hamilton (2016), Mascarenhas, Kesavan & Bernacchi (2006), Shaw (2004), Schouten, McAlexander & Koenig (2007), Rusu et al. (2018), Zhang (2014), Shaw, Dibeehi & Walden (2010), Yang, Yang & Wen (2010), Chang & Lin (2015), Watkinson (2013)* | 7 | *Gentile, Spiller & Noci (2007), Becker & Jaakkola (2020), Shaw & Ivens (2002), De Keyser et al. (2015), Klaus (2015), Knutson & Beck (2007), Kim et al. (2011)* |
| Healthcare[3] | 2 | *Klaus (2018)* | 1 | *Deshwal & Bhuyan (2018)* |
| Hotel | 1 | *Peng, Zhao & Mattila (2015)* | 8 | *Zhou & Mu (2013), Walls (2013), Nicholas & Lee (2017), Ali, Hussain & Ragavan (2014), Ren et al. (2016), Walls et al. (2011), Rageh, Melewar & Woodside (2013), Knutson et al. (2009)* |
| Management | 4 | *Johnston & Clark (2008), Buttle & Maklan (2015), Benzarti & Mili (2018), Homburg, Jozić & Kuehnl (2017)* | – | – |
| Market[4] | 2 | *Beaudon & Soulier (2019), Boakye, Chiang & Tang (2016)* | 2 | *Shaw & Ivens (2002)* |
| Marketing[5] | 3 | *Schmitt (1999), Lemon & Verhoef (2016)* | 3 | *Schmitt (1999), Lemon & Verhoef (2016), Klaus (2011)* |
| Mass-Catering Service | – | – | 1 | *Maslov (2019)* |
| Mortgage | 1 | *Klaus & Maklan (2013)* | 1 | *Klaus & Maklan (2013)* |
| New Technology | – | – | 1 | *Hoyer et al. (2020)* |
| Non-trading Virtual Community | – | – | 1 | *Tu & Zhang (2013)* |
| Online Product Communities[6] | – | – | 1 | *Nambisan & Watt (2011), Salehi, Salimi & Haque (2013)* |
| Organization | – | – | 1 | *Yang & Wang (2010)* |

(Continued)

| Domain | CX definitions | | | CX dimensions | |
|---|---|---|---|---|---|
| | Number of definitions | References | | Number of dimension proposals | References |
| P2P Accommodation | – | – | | 1 | *Lyu, Li & Law (2019)* |
| Restaurant | 1 | *Walter, Edvardsson & Öström (2010)* | | – | – |
| Retail | 10 | *Verhoef et al. (2009), Klaus & Maklan (2013), Zhao & Deng (2020), Tran, Tuyet & Hara (2017), Roy et al. (2017), Khan et al. (2020), Bagdare & Jain (2013), Bleier, Harmeling & Palmatier (2019), Anninou & Foxall (2019), Pekovic & Rolland (2020)* | | 20 | *Verhoef et al. (2009), Srivastava & Kaul (2016), Klaus & Maklan (2013), Shaw & Ivens (2002), Zhao & Deng (2020), Tran, Tuyet & Hara (2017), Roy et al. (2017), Bagdare & Jain (2013), Bleier, Harmeling & Palmatier (2019), Pekovic & Rolland (2020), Grewal, Levy & Kumar (2009), Shi et al. (2020), Rose et al. (2012), Kumar & Anjaly (2017), Zaharia & Schmitz (2020), Sathish & Ganesan (2015), Roy, Gruner & Guo (2020), Pandey & Chawla (2018), Singh (2019), Peltola, Vainio & Nieminen (2015)* |
| Service Encounter | 1 | *Larivière et al. (2017)* | | – | – |
| Smart Service | – | – | | 1 | *Gonçalves et al. (2020)* |
| Social Commerce | – | – | | 1 | *Zhang et al. (2014)* |
| Sport | 2 | *Yoshida (2017), Klaus & Maklan (2011)* | | 1 | *Klaus & Maklan (2011)* |
| Sustainability | 1 | *Signori et al. (2019)* | | – | – |
| Telecommunication | 5 | *Belabbes & Oubrich (2018), Sirapracha & Tocquer (2012), Menachem et al. (2015), Joshi (2014), Chen et al. (2012)* | | 6 | *Joshi (2014), Belabbes & Oubrich (2018), Belabbes, Aziza & Mourad (2017), Chen et al. (2012), Sujata et al. (2016), Menachem et al. (2015)* |
| Theme Park | 1 | *Ali et al. (2018)* | | 1 | *Ali et al. (2018)* |
| Tourism | 1 | *Gopalan & Narayan (2010)* | | – | – |
| Transport | – | – | | 1 | *Pareigis, Edvardsson & Enquist (2011)* |

**Notes:**
[1] Studies *Klaus & Maklan (2012)* and *Klaus & Maklan (2013)* present exactly the same definition for the Banks domain, for this reason the number of definitions is three (and not four). At the same time, studies *Klaus & Maklan (2012), Klaus & Maklan (2013)*, and *Maklan & Klaus (2011)* present exactly the same dimensions for Banks, for this reason the number of proposed dimensions is nine (and not 11).
[2] Studies *Pei et al. (2015a and 2015b)* present exactly the same definition for the E-Commerce domain, for this reason the number of definitions is three (and not four).
[3] The study *Klaus (2018)* presents two different definitions for the Health-care domain, for this reason the number of definitions is two (and not one).
[4] The study *Shaw & Ivens (2002)* presents two different proposals of dimensions for the Market domain, for this reason the number of proposed dimensions is three (and not two).
[5] The study *Lemon & Verhoef (2016)* presents two different definitions for the Marketing domain, for this reason the number of definitions is three (and not two).
[6] Studies *Nambisan & Watt (2011)* and *Salehi, Salimi & Haque (2013)* present exactly the same dimensions for the Online Product Communities domain, for this reason the number of dimensions proposed is one (and not two).

(see Table 8). Specifically, the domains with the highest number of proposed CX definitions are retail (10 definitions proposed), telecommunication (five definitions proposed), and management (four definitions proposed). Most of the CX dimensions are proposed in the contexts of Retail (20 proposals), Banks (11 proposals), and Hotels (eight proposals). For instance, for the retail domain, 20-dimensional proposals are developed. Of these, seven proposals are aimed at describing the CX dimensions in Retail in a general way, while three-dimensional proposals have been proposed for omnichannel retail, three for online retail, and one proposal for post-purchase retail, online grocery, traditional and modern markets, smart retail, luxury retail, clothing e-retail, and DIY retailing sectors.

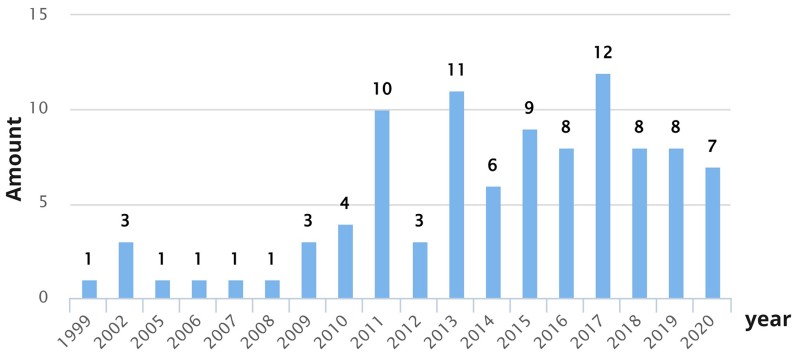

**Figure 8  Number of domains studied per year.**

Figure 8 shows the number of domains studied per year, considering studies that have proposed both definitions and dimensions for the CX. As shown in Fig. 8, over the years the domains in which the CX is studied have increased. Over the years, the number of domains studied has been increasing. In 2013 and 2017, 11 and 12 different domains were studied, respectively. In 2013, the domains explored were Bank, Casino, E-commerce, Fuel and service station, General, Hotel, Mortgage, Non-trading Virtual Community, Online Product Communities, Restaurant, and Retail, while in 2017, the domains explored were Airline, Bank, Brand, E-commerce, Food, and wine event, General, Hotel, Management, Retail, Service encounter, Sport, and Telecommunications.

## Relation between customer experience and human-computer interaction (RQ4)

No studies were found that propose CX definitions or dimensions directly related to HCI. However, several authors include certain characteristics of HCI in their proposals. Table 9 presents the definitions that present aspects related to HCI, the proposals containing dimensions related to HCI, and the different domains that cover topics related to the HCI field.

   Fourteen CX definitions that highlight aspects of HCI were identified (see Table 9). For instance, some definitions mention different concepts related to HCI, such as "products", "technologies", "media and social networks", "websites", "software systems" or "digital interfaces". These aspects can be observed in 10 different domains: Telecommunication, E-commerce, Blog, Management, Market, Retail, Sport, Service encounter, General, and Business. In addition, some of these definitions refer directly to the concept of HCI. *Liu et al. (2017)* mentioned that "customer experience is reflection and feeling during the process of interaction with enterprises under the situation of human-computer interaction", while *Rusu et al. (2018)* commented that "customer experience (CX) is traditionally related to the Service Science field. It is not limited to the user,—(software) product interaction but refers to the whole customer and—company (or companies) interaction through several products. When some of these products are software systems, the HCI interest in CX becomes obvious".

**Table 9  CX definitions, CX dimension proposals, and domains that cover HCI aspects.**

| | Amount | References |
|---|---|---|
| CX definitions that highlight HCI aspects | 14 | *Sirapracha & Tocquer (2012), Klaus (2013), Pei et al. (2015b), Chen & Lin (2015), Buttle & Maklan (2015), Boakye, Chiang & Tang (2016), Roy et al. (2017), Yoshida (2017), Larivière et al. (2017), Liu et al. (2017), Rusu et al. (2018), Behare, Waghulkar & Shah (2018), Bleier, Harmeling & Palmatier (2019), Pekovic & Rolland (2020)* |
| CX dimensions proposal that contains HCI aspects or dimensions | 24 | *Shaw & Ivens (2002), Hsu & Tsou (2011), Nambisan & Watt (2011), Rose et al. (2012), Salehi, Salimi & Haque (2013), Klaus (2013), Tu & Zhang (2013), Pei et al. (2015a, 2015b), Parise, Guinan & Kafka (2016), Loureiro & Sarmento (2017), Kumar & Anjaly (2017), Roy et al. (2017), Pandey & Chawla (2018), Su et al. (2019), Bleier, Harmeling & Palmatier (2019), Shi et al. (2020), Hoyer et al. (2020), Pekovic & Rolland (2020), Zaharia & Schmitz (2020), Shin, Cho & Lee (2019), Zhao & Deng (2020), Saberian et al. (2020), Gonçalves et al. (2020)* |
| Domains that cover topics related to the HCI field | 18 | Telecommunication: *Sirapracha & Tocquer (2012)*; E-Commerce: *Klaus (2013), Pei et al. (2015b), Liu et al. (2017), Pei et al. (2015a)*; Blog: *Chen & Lin (2015), Hsu & Tsou (2011)*; Management: *Buttle & Maklan (2015)*; Market: *Boakye, Chiang & Tang (2016), Shaw & Ivens (2002)*; Retail: *Rose et al. (2012), Roy et al. (2017), Bleier, Harmeling & Palmatier (2019), Pekovic & Rolland (2020), Kumar & Anjaly (2017), Pandey & Chawla (2018), Bleier, Harmeling & Palmatier (2019), Shi et al. (2020), Zaharia & Schmitz (2020), Zhao & Deng (2020)*; Sport: *Yoshida (2017)*; Service Encounter: *Larivière et al. (2017)*; General: *Rusu et al. (2018)*; Business: *Behare, Waghulkar & Shah (2018)*; Online product communities: *Nambisan & Watt (2011), Salehi, Salimi & Haque (2013)*; Non-trading virtual community: *Tu & Zhang (2013)*; Digital technology: *Parise, Guinan & Kafka (2016)*; Bank: *Loureiro & Sarmento (2017), Shin, Cho & Lee (2019)*; Fresh products APPs: *Su et al. (2019)*; New technologies: *Hoyer et al. (2020)*; Digital platforms: *Saberian et al. (2020)*; Smart service: *Gonçalves et al. (2020)* |

Regarding the dimensions, 24 proposals contained dimensions that could be related to HCI, such as accessibility, usability, speed, interactivity, and personalization (see Table 9). For instance, *Nambisan & Watt (2011)* proposed four dimensions where three could be linked to HCI (*i.e.*, pragmatic experience, hedonic experience, and usability experience). Similarly, two of the five dimensions mentioned by *Pei et al. (2015a)* could be associated with HCI (*i.e.*, website usefulness, and website ease of use). In addition, there are proposals where all their dimensions apply to HCI (*Rose et al., 2012; Parise, Guinan & Kafka, 2016; Pandey & Chawla, 2018; Su et al., 2019; Saberian et al., 2020; Zhao & Deng, 2020*). Finally, these dimensions were presented in 12 different domains, *i.e.*, Market, Blog, Online product communities, Retail, E-Commerce, Non-trading virtual community, Telecommunication, Digital technology, Bank, Fresh products APPs, New technologies, Digital platforms, and Smart service.

Figure 9 shows CX definitions and CX dimension proposals that present aspects related to HCI over the years. In 2002, *Shaw & Ivens (2002)* proposed CX dimensions that include some aspects of HCI (accessibility, user-friendliness, and excitement). As shown in Fig. 9, the years with the largest number of publications correspond to 2020 (seven studies that cover HCI aspects related to CX definitions) and 2017 (four studies that cover HCI aspects related to CX definitions and three studies that cover HCI aspects related to CX dimensions). It is possible to observe that over the years, interest in investigating CX from an HCI perspective has increased, specifically since 2011. Researchers are progressively more concerned about studying the CX from an HCI point of view.

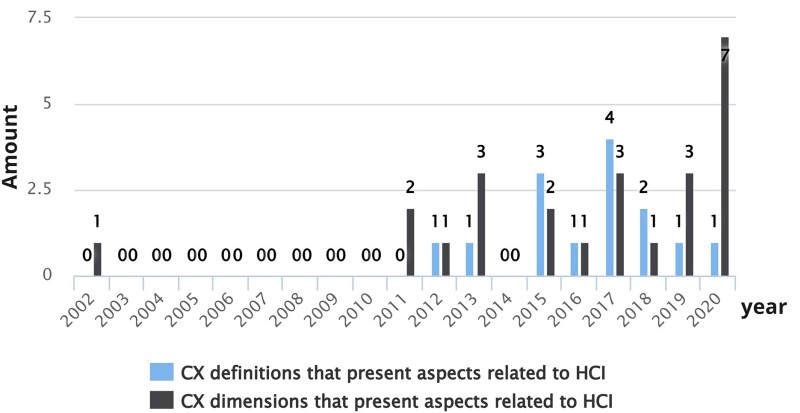

**Figure 9 Number of CX definitions and dimensions per year that present aspects related to HCI.**

## DISCUSSION

In this section, the main review findings are discussed and a detailed analysis regarding the research questions, highlighting the relation between CX and HCI, is provided. In addition, a novel CX definition and dimensions from an HCI perspective and recommendations for creating new CX dimensions in domains related to HCI are proposed.

### About customer experience definitions and domains (RQ1 and RQ3)

As presented in the Results section, the most cited CX definitions correspond to those proposed by *Meyer & Schwager (2007)*, *Verhoef et al. (2009)*, *Gentile, Spiller & Noci (2007)*, *Schmitt (1999)*, *Lemke, Clark & Wilson (2011)*, and *Lemon & Verhoef (2016)*. Table 10 shows a comparative summary of the different proposed definitions. These CX definitions were proposed for a general domain, as well as specific domains, such as retail, business, and marketing.

The definitions proposed by *Meyer & Schwager (2007)* and *Lemke, Clark & Wilson (2011)* have several similarities. Both proposals emphasize customer responses as a result of direct or indirect contact with a company (Meyer and Schwager proposal) or firm (Lemke et al. proposal). *Verhoef et al. (2009)* also emphasize customer responses, but the authors indicate that this experience can be created both by elements that the retailer can control and by elements that are beyond their control. Furthermore, in 2016 *Lemon & Verhoef (2016)* complemented the definition proposed by *Verhoef et al. (2009)* in 2009, indicating that the CX is a multidimensional construct and incorporating new CX dimensions into the definition. On the other hand, *Gentile, Spiller & Noci (2007)* highlight that the CX is strictly personal, and that this experience arises from a set of interactions between the customer and a product or company, while *Schmitt (1999)* indicates that the CX occurs as a result of encountering, undergoing, or living through things.

When reviewing the proposals of different authors, many of them incorporate several dimensions to describe or characterize the CX within the definition. Regarding the most cited CX definitions, four incorporate dimensions into their definition proposal (*Verhoef*

**Table 10 Comparative summary between the most cited CX definitions.**

| | CX definition proposals | | | | | |
| --- | --- | --- | --- | --- | --- | --- |
| | *Meyer & Schwager (2007)* | *Verhoef et al. (2009)* | *Gentile, Spiller & Noci (2007)* | *Schmitt (1999)* | *Lemke, Clark & Wilson (2011)* | *Lemon & Verhoef (2016)* |
| Domain | General | Retail | General | Marketing | Business | Marketing |
| Main concepts | Subjective response; Direct and indirect contact; Purchase, use | Response; Experience; Purchase and consumption; Holistic view | Experience; Set of interactions | Experience; Value | Subjective response; Direct and indirect encounter; Holistic view | Response; Multidimensional view; Purchase journey |
| Definition approach | Responses and encounters | Responses and elements that the retailer can and cannot control | Interactions between the customer and a company | Experiences as a result of encounters | Responses and encounters | Responses and customer journey |
| The customer is related to a | Company | Retailer | Company | N/A | Firm | Firm |
| Do they include CX dimensions in the definition? | No | Yes | Yes | Yes | No | Yes |

*et al., 2009*; *Gentile, Spiller & Noci, 2007*; *Schmitt, 1999*; *Lemon & Verhoef, 2016*). Specifically, these four CX definitions incorporate the "emotional" dimension into the description (the dimension related to the generation of moods, feelings, and emotions that create an affective experience), while *Verhoef et al. (2009)*, *Schmitt (1999)*, and *Lemon & Verhoef (2016)* incorporate the "cognitive" dimension; and *Gentile, Spiller & Noci (2007)*, *Schmitt (1999)*, and *Lemon & Verhoef (2016)* incorporate the "sensorial" dimension. Other CX dimensions that are incorporated in the CX definitions are "affective", "social", "physical", "rational", "spiritual", and "behavioral".

In this literature review, the first CX definition corresponds to the one proposed by *Schmitt (1999)* in 1999. This definition was proposed in the Marketing domain, emphasizing the experiences that occur as a result of an encounter. In addition, the author incorporates CX dimensions that provide value. When comparing this first proposal with the latest definitions found in the review (2020), it is still common to incorporate dimensions within the CX definition (*Khan et al., 2020*; *Pekovic & Rolland, 2020*) and the concept of "value" is still relevant (*De Keyser et al., 2020*). In addition, new definitions have been proposed for different domains, including their specific characteristics and highlighting elements such as the holistic view of the CX and its multidimensionality. On the other hand, various authors add concepts, such as "touchpoints", to explain the moments in which the customer interacts with a brand or company, and "channels" to explain the method ("medium" or "ways") by which the customer interacts with the brand or company (*e.g.*, websites, emails, phone calls, social networks, among others).

## About customer experience dimensions and domains (RQ2 and RQ3)

Figure 10 shows the number of CX dimensions present in the different proposals identified. The most frequent amount of CX dimensions within a proposal varied between

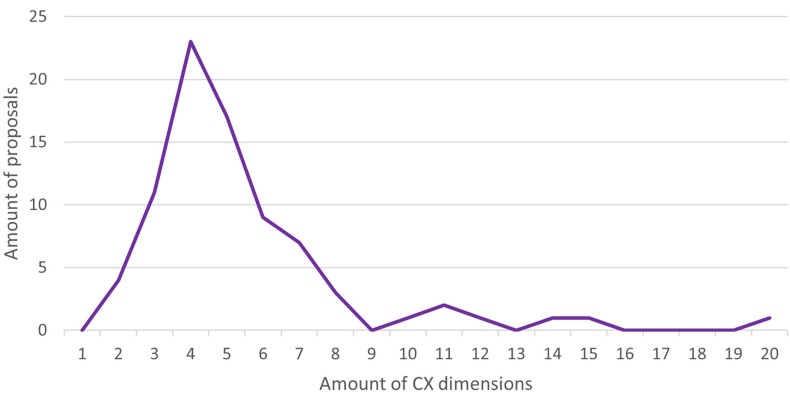

**Figure 10** **Amount of CX dimensions per proposal.**

three and six dimensions (60 of 81 proposals, 74.1%). The most popular number of dimensions was four (23 of 81 proposals, 28.4%). In addition, the lowest number of dimensions among the studies was two (four proposals), such as "emotional experience" and "relationship experience" for the non-trading virtual community domain (*Tu & Zhang, 2013*), or "hedonic experience" and "cognitive experience" for the digital platform's domain (*Saberian et al., 2020*). In contrast, the highest amount found was 20 dimensions (one proposal) (*Joshi, 2014*) suggesting dimensions such as "network coverage", "complaint handling", "marketing communication", and "billing transparency" for the Telecommunication domain.

After analyzing and discussing all the proposals identified (81 proposals), the ideal number of CX dimensions within a proposal should be between four and six since (1) proposals with few dimensions (less than four) may be too general, ignoring important CX elements, and (2) proposals with many dimensions (more than six) may be too specific and can only be used for the specific domain or even lead to confusion. Proposals with four to six dimensions may be easier to apply and/or adapt for the same domain, as for other domains.

The dimension proposals are usually represented in different ways, such as text, figures (diagrams, models, or frameworks), and tables. It is common for authors to use more than one way to represent dimensions to facilitate their understanding by mixing text, tables, and/or figures. However, it was observed that, although the studies included text, figures, or tables to present the proposal, approximately half of these (36 of 81 proposals, 44.4%) did not provide a definition to explain the dimensions. This lack of definition and/or explanation of the CX dimensions makes it difficult to understand the proposals, complicating the application of these studies in the future (either in the same domain or similar) since they could be easily misunderstood or misinterpreted by other researchers.

Furthermore, some proposals group the dimensions into different categories. Nevertheless, this is not common in the studies reviewed, since only 17.3% of the proposals (14 of 81) present the dimensions grouped into categories. For instance, *Pandey & Chawla (2018)* propose the categories "functionality factors" and "psychological factors" to group 10 dimensions in the retail domain; *Wasan (2018)* uses "functional clues", "mechanic

**Table 11 Similarity between CX dimensions proposed in most cited studies.**

| Schmitt (1999) | Gentile, Spiller & Noci (2007) | Brakus, Schmitt & Zarantonello (2009) |
|---|---|---|
| Sense | Sensorial | Sensory |
| Think | Cognitive | Intellectual |
| Act | Lifestyle | Behavioral |
| Feel | Emotional | – |
| Relate | Relational | – |
| – | Pragmatic | – |
| – | – | Affective |

clues", and "humanic clues" to group six dimensions in the bank domain, and *Wong & Wu (2013)* use the categories "service environment", "employee service", "value", "hedonic and novelty", "brand experience", and "perceived luck" to group 15 dimensions in the casino domain. Although using categories is not essential, using categories to group dimensions can help in understanding them. In addition, these categories can be used as a reference to include or propose new dimensions for other domains.

Regarding the number of citations, the most cited CX dimension proposals correspond to those presented by *Schmitt (1999)*, *Gentile, Spiller & Noci (2007)*, *Verhoef et al. (2009)*, and *Brakus, Schmitt & Zarantonello (2009)*. These dimensions were proposed for the Marketing, General, Retail, and Brand domains. These proposals include some of the most commonly used CX dimensions (see "Proposals for Customer Experience Dimensions (RQ2)", Table 7), such as "cognitive", "emotional", "sense", "relate", "affective", and "behavioral" (see Table 11 below). In addition, although the proposal of *Verhoef et al. (2009)* is widely cited as a dimension, it corresponds to determinants or antecedents for the creation of CX, and the dimensions proposed are those mentioned in their CX definition (*i.e.*, "cognitive", "affective", "social", "physical", and "emotional").

It was distinguished that the authors share some dimensions in their proposals (see Table 11), however, they use synonyms to present them. For instance, *Schmitt (1999)* proposes the "sense" dimension, while *Gentile, Spiller & Noci (2007)* mention "sensorial", and *Brakus, Schmitt & Zarantonello (2009)* call it "sensory". This may be explained by the fact that these two studies—like other proposals—use Schmitt's proposal as a basis for their dimensions, including new ones adapted to other domains, as is the case of the "pragmatic" proposed by *Gentile, Spiller & Noci (2007)* or "affective" by *Brakus, Schmitt & Zarantonello (2009)*.

When analyzing the proposals of CX dimensions of the most studied domains (*i.e.*, Retail, Bank, and Hotel) some CX dimensions that are mentioned in more than one proposal were detected. For instance, in the retail domain, the dimensions used in more than one proposal are: "price", "feel", "relate", "sense", "cognitive", "emotional", "social", "personalization", "informativeness", and "interactivity", while in the bank domain are "convenience", "moments-of-truth", and "customization".

Compared with retail and bank domains, in the hotel domain, the same CX dimensions are not used in more than one proposal, despite being one of the most studied domains. It seems possible that these results are because the Hotels domain is considered too broad or complex to analyze, so the authors have performed studies for specific subdomains, such as Budget hotels (*Ren et al., 2016*), Resort hotels (*Ali, Hussain & Ragavan, 2014*), or Luxury hotels (*Walls et al., 2011*). Due to the above, the identified proposals are usually more specific and consider the characteristics of the specific domain analyzed.

Another interesting finding is related to the CX dimensions proposed for general domains (*i.e.*, proposals that can be used for any domain). We noticed that seven proposals can be applied to describing the CX in any domain (*Gentile, Spiller & Noci, 2007*; *Becker & Jaakkola, 2020*; *Shaw & Ivens, 2002*; *De Keyser et al., 2015*; *Klaus, 2015*; *Knutson & Beck, 2007*; *Kim et al., 2011*). In addition, some of these seven proposals were created based on the most cited studies related to CX dimensions (*Schmitt, 1999*; *Gentile, Spiller & Noci, 2007*; *Verhoef et al., 2009*; *Brakus, Schmitt & Zarantonello, 2009*). For instance, *De Keyser et al. (2015)* used the dimensions proposed by *Gentile, Spiller & Noci (2007)*, and *Brakus, Schmitt & Zarantonello (2009)* as a reference. In contrast, *Becker & Jaakkola (2020)* propose dimensions based on *Schmitt (1999)*, *Verhoef et al. (2009)*, and *Lemon & Verhoef (2016)*. Among these seven proposals, there are certain common dimensions, which are "accessibility", "cognitive", "affective", "emotional", "environment", and "sensorial". Variations in dimensions were also observed, such as "physical" and "physical response" or "social" and "social response".

After conducting this review, the CX dimensions proposed by *Gentile, Spiller & Noci (2007)* (*i.e.*, "sensory", "emotional", "cognitive", "pragmatic", "lifestyle", and "relational") are considered one of the most complete since (1) it includes essential CX dimensions to evaluate the CX in different interactions throughout the customer journey; (2) it is a general proposal of dimensions that can be applied—and if necessary adapted—in different domains without major drawbacks; and (3) its dimensions are appropriately defined/explained, including examples to better understanding.

## About customer experience and the relation with human-computer interaction (RQ4)

As mentioned in "Relation Between Customer Experience and Human-Computer Interaction (RQ4)", no studies were found that propose a CX definition or CX dimensions from an HCI perspective. However, it is important to highlight that 18 domains were found that contained CX definitions and/or dimensions that could be related to HCI (see Tables 9 and 12). The domains that highlight the most aspects of HCI in the CX definitions corresponded to retail (three definitions) and e-commerce (three definitions). The domains that present more CX dimension proposals related to HCI are retail (nine proposals) and e-commerce (three proposals). In contrast, the domains that present aspects related to HCI both in definitions and dimensions are retail, e-commerce, blog, and market.

On the other hand, five studies present CX definitions and CX dimensions that could be related to HCI in the same article. Concepts such as mental perception, transactions, or

**Table 12 Number of studies that cover HCI aspects per domain.**

| Domains | Number of studies | CX definitions that cover aspects related to HCI | CX dimension proposals that cover aspects related to HCI |
|---|---|---|---|
| 1. Retail | 10 | 3 | 9 |
| 2. E-Commerce | 4 | 3 | 3 |
| 3. Blog | 2 | 1 | 1 |
| 4. Market | 2 | 1 | 1 |
| 5. Online product communities | 2 | 0 | 2 |
| 6. Bank | 2 | 0 | 2 |
| 7. Telecommunication | 1 | 1 | 0 |
| 8. Management | 1 | 1 | 0 |
| 9. Sport | 1 | 1 | 0 |
| 10. Service encounter | 1 | 1 | 0 |
| 11. General domain | 1 | 1 | 0 |
| 12. Business | 1 | 1 | 0 |
| 13. Non-trading virtual community | 1 | 0 | 1 |
| 14. Digital technology | 1 | 0 | 1 |
| 15. Fresh products apps | 1 | 0 | 1 |
| 16. New technologies | 1 | 0 | 1 |
| 17. Digital platforms | 1 | 0 | 1 |
| 18. Smart service | 1 | 0 | 1 |
| TOTAL | 34 | 14 | 24 |

technology were identified in CX definitions. Likewise, dimensions such as usability, interactivity, and entertainment were suggested in CX dimension proposals. Table 13 shows the key concepts and dimensions detected in these studies related to HCI and grouped into the HCI aspects. Most of these concepts and dimensions were grouped into the aspect of "human characteristics" since CX is mainly related to characteristics such as cognition, feelings, or behavior. In contrast, only one element was associated with the aspect "software product development process" (usability). Due to the lack of elements related to this HCI aspect, it is recommended to consider it when proposing definitions and/or dimensions. None of the key concepts and dimensions detected were grouped into the dimension "nature of human-computer interaction" because the proposals were focused on retail or e-commerce domains instead of HCI.

CX dimensions proposed in the 24 studies that contain HCI aspects were analyzed (see "Relation Between Customer Experience and Human-Computer Interaction (RQ4)", Table 9) and the most commonly used dimensions were detected (see Table 14). Each dimension presented may have different variations; for example, the "customer" dimension includes alternatives, such as "customer support", "customer characteristics", "customer participation and interaction", and "employee-customer engagement". The

**Table 13 Studies that present CX definitions and dimensions that cover HCI aspects.**

| Authors | Domain | Concepts and dimensions that stand out per each HCI aspect | | | |
|---------|--------|----------------------|--------------------|-----------------------------|------------------------------------|
| | | Use and context of system | Human characteristics | System and interface architecture | Software product development process |
| *Klaus (2013)* | E-Commerce | Interactivity, product presence | Psychological features, mental perceptions, behavior, social elements, emotions | Functionalities | Usability |
| *Pei et al. (2015b)* | E-Commerce | Interactive service, ease of use | Feelings, cognitive elements, psychological features | Shopping conditions, transactions | None |
| *Roy et al. (2017)* | Retail | Interactivity, Personalization | Perceived enjoyment and control | Technology | None |
| *Bleier, Harmeling & Palmatier (2019)* | Retail | Sensory, entertainment | Psychological features, cognitive elements, social elements | None | None |
| *Pekovic & Rolland (2020)* | Retail | Sensorial | Social, cognitive elements, behavior | Technology responses | None |

**Table 14 Most used CX dimensions in studies that cover HCI aspects.**

| No. | Dimension | Amount | Nature of HCI | Use and context of system | Human characteristics | Computer system and interface architecture | Software development process |
|-----|-----------|--------|---------------|---------------------------|----------------------|-------------------------------------------|------------------------------|
| 1 | Social | 7 | | x | x | | |
| 2 | Cognitive | 6 | | | x | | |
| 3 | Emotional | 5 | | | x | | |
| 4 | Sensorial | 4 | | x | | x | |
| 5 | Customer | 4 | | | x | | |
| 6 | Ease | 4 | | x | | x | |
| 7 | Hedonic | 3 | | | x | | |
| 8 | Usability | 3 | | x | | x | x |
| 9 | Affective | 3 | | | x | | |
| 10 | Interactivity | 3 | | x | | x | |
| 11 | Trust | 3 | | x | | | |
| | | **Total** | 0 | 6 | 6 | 4 | 1 |

dimensions most commonly used in the 24 proposals examined are "social" (seven times), "cognitive" (six times), and "emotional" (five times).

Each CX dimension with the HCI aspect related was matched (see Table 14). The HCI aspects most covered in the 24 studies were "use and context of system" (six dimensions), "human characteristics" (six dimensions), and "computer system and interface architecture" (four dimensions). Unexpectedly, the "software development process" HCI aspect was covered by one dimension (usability), highlighting the need to include a dimension within a new HCI-dimension proposal to cover this aspect. The aspect "nature

of human-computer interaction" was not covered by any dimension since the proposals did not focus on HCI.

## Overview of systematic literature review findings

A total of 122 studies that proposed definitions and/or dimensions for CX in different domains throughout the years were identified. Among these articles, some stand out as they have been widely cited in the literature (see Tables 5 and 6). This may be because these studies (1) are the first conducted in the CX area (*Schmitt, 1999*); (2) propose definitions and/or dimensions for domains where other researchers are also working (*Verhoef et al., 2009*; *Gentile, Spiller & Noci, 2007*); (3) propose both definitions and dimensions in the same study (*Schmitt, 1999*; *Verhoef et al., 2009*; *Gentile, Spiller & Noci, 2007*); or (4) update their previous proposals with new and novel aspects (*Lemon & Verhoef, 2016*).

Years were identified where CX definitions were proposed but not CX dimensions, such as 2005 and 2006. In addition, different years where no proposals were found were detected, such as 2000, 2001, 2003, and 2004. This result may be because the review mainly includes articles since 2010, and those obtained before that year were only included using the backward snowballing method. Using this method, the oldest study found proposing a CX definition was in 1999 (also the oldest that proposes CX dimensions).

One of the most cited articles found in this review is the study performed by *Schmitt (1999)* in the Marketing domain, as it is one of the first articles (1999) about CX in the literature and is widely used as a basis in multiple studies (28 citations, 10 for its definition and 18 for its dimensions). In addition, the study conducted by *Gentile, Spiller & Noci (2007)* is also widely cited, as it can be easily applied and adapted for any domain, which makes it a very complete article within the literature (23 citations, 11 for its definition and 12 for its dimensions). The study performed by *Verhoef et al. (2009)* also has several citations, as they proposed both a definition and dimensions in their study, including a conceptual model for CX creation in the retail domain (21 citations, 12 for its definition and nine for its dimensions). Finally, the study conducted by *Meyer & Schwager (2007)* is also widely cited, as they provided novel aspects within the proposal making it the most cited definition identified in this review.

One interesting finding is that 27 studies included both CX definitions and CX dimensions in the same article. The oldest proposal found was the study conducted by *Schmitt (1999)* in 1999. Surprisingly, the studies that propose CX definition and CX dimensions in the same article are also some of the most cited in the literature (*Schmitt, 1999*; *Gentile, Spiller & Noci, 2007*; *Verhoef et al., 2009*).

Although only 14 CX definitions highlight aspects of HCI, most of the CX definitions identified in the systematic literature review could be related to HCI as they focus their definitions on different moments of contact or touchpoints such as *Gentile, Spiller & Noci (2007)*, *Shaw & Ivens (2002)*, *Joshi (2014)*, and *Lemon & Verhoef (2016)*. This is because each of these moments of contact or touchpoints that arise in the customer journey along the prepurchase, purchase, and post-purchase phases may be related to software systems with which customers interact. Therefore, domains such as Bank, Tourism, Restaurants, Automobile, airlines, Hotels, Health-care, Theme parks, and Education could also be

## OVERVIEW OF THE SLR FINDINGS

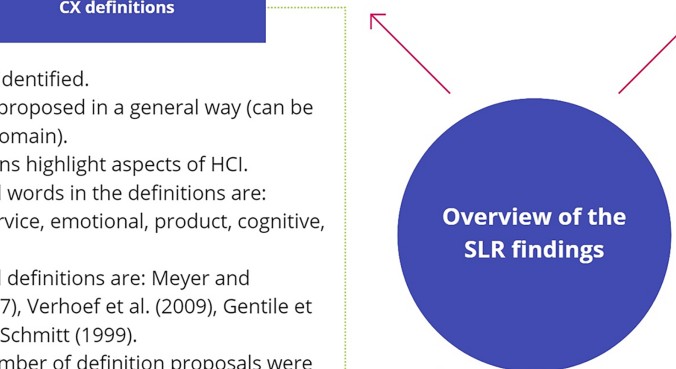

**CX definitions**

- 71 definitions identified.
- 23 definitions proposed in a general way (can be used for any domain).
- 14 CX definitions highlight aspects of HCI.
- The most used words in the definitions are: interaction, service, emotional, product, cognitive, and response.
- The most cited definitions are: Meyer and Schwager (2007), Verhoef et al. (2009), Gentile et al. (2007), and Schmitt (1999).
- The largest number of definition proposals were published in 2018 (10 studies).

**CX dimensions**

- 81 dimension proposals identified.
- 20 dimension proposals have been created in the Retail domain.
- 24 proposals contained dimensions that could be related to HCI.
- The most used dimensions are: social, cognitive, emotional, environment, and sense.
- The most cited dimension proposals are: Schmitt (1999), Gentile et al. (2007), Verhoef et al. (2009), and Brakus et al. (2009).
- The largest number of definition proposals were published in 2020 (11 studies).

**Overview of the SLR findings**

**CX application domains**

- 39 main domains identified.
- 79 specific domains identified.
- 18 domains contain CX definitions and/or dimensions that could be related to HCI.
- The main domains with the most proposed CX definitions are: Retail, Telecommunication, and Management.
- The domains with the most proposed CX dimension proposals are: Retail, Bank, and Hotel.
- In 2013 and 2017 were studied the largest number of domains (11 different domains in 2013 and 12 in 2017).
- The Retail domain presents the largest number of definitions and dimension proposals (10 definitions, 20 dimension proposals).

**SLR results**

- 122 studies reviewed and analyzed.
- 27 studies include CX definitions and CX dimensions in the same article.
- 5 studies propose CX definitions and CX dimensions that could be related to HCI in the same article.
- Since 2015, the number of studies proposing CX definitions has increased.
- Since 2011, the number of studies proposing CX dimensions has increased.
- 66.2% of CX definitions were published in Journals and 18.3% in conferences.
- 64.8% of CX definitions were detected using the queries; 35.2% were detected through snowball.

- 73.4% of CX dimension proposals were published in Journals and 21.5% in conferences.
- 69.6% of CX dimension proposals were detected using the queries; 30.4% were detected through snowball.
- 74.1% of the proposals include between 3 and 6 dimensions. Most of the studies propose 4 dimensions (28.4%).
- 44.4% of the studies do not explain its dimensions (authors do not include a definition).
- 17.3% of the studies group their dimensions into categories.

**Figure 11 Overview of the systematic literature review findings.**

included. Regarding CX dimension proposals, several CX dimensions could be indirectly associated with HCI. For instance, emotional dimensions, dimensions of interaction with staff, dimensions of perception of the environment, and dimensions of trust, security, and reliability. Therefore, several domains, such as Brands, Hotels, Theme parks, casinos, and Health-care could also be included.

Figure 11 illustrates the main findings related to each research question (RQ1: CX definitions, RQ2: CX dimensions, RQ3: CX application domains, and RQ4: relation between CX and HCI), along with the main results of the systematic literature review (SLR).

## A novel customer experience definition and dimensions for human-computer interaction

The CX is an important aspect for a wide range of scientists, academics, and companies. Interestingly, this is noticeable in this literature review, with the large number of studies that have been performed. Numerous terms are used to describe CX, but there is no formal definition from the point of view of HCI. Therefore, and based on the studies analyzed in this review, the following general definition of the CX from the point of view of HCI is proposed. This definition considers the most used and relevant concepts, adding new elements that are necessary to understand its multidimensional and holistic vision:

"**The customer experience corresponds to the sum of the experiences of a customer when interacting with different products, systems, or services offered by a brand, company, or organization over time. It emerges from a customer journey that consists of three distinct yet interconnected stages: pre-interaction, interaction, and post-interaction.** The experience at each moment of interaction should be meaningful and valuable to the customer. This experience is built based on the customer's perceptions, responses, and emotions when they are not yet interacting directly with a product, system, or service offered by a company (pre-interaction); when they are interacting with any of their products, systems, or services (interaction); and when they are no longer interacting with the products, systems, or services (post-interaction). All these interactions (pre-interaction, interaction, and post-interaction)—whether direct or indirect—conform to the customer journey, which is made up of different touchpoints (moments of interaction). The customer can interact with the brand or company through different channels, and the customer experience can be different at the same touchpoint depending on the channel used".

On the other hand, the following five general dimensions to describe CX from an HCI perspective are proposed (see Fig. 12): social, cognitive, emotional, sensorial, and software product dimensions. The process used to create this proposal is shown in Fig. 13 (the specific activities are proposed in "Implications, Novelty, and Recommendations for Proposing New CX Dimensions"). Although the validation and refinement of this proposal have not yet been performed, the dimensions are appropriate and related to CX and HCI, since the creation of this model is based on (1) the 81 proposed dimensions analyzed in this systematic literature review, (2) the 24 dimension proposals that contain HCI aspects (see "About Customer Experience and the Relation with Human-COmputer Interaction (RQ4)"), and (3) the four aspects of HCI (*Hewett et al., 1992*) (see "Relationship Between Human-Computer Interaction, Customer Experience, and User Experience"): "use and context of software", "human characteristics", "system and interface architecture", and "software product development process". The "nature of human-computer interaction" aspect was not included since it relates to the domain in which HCI is studied. In addition, the five dimensions are included as they cover the most relevant aspects of HCI (human characteristics and elements of software products) along with the most distinctive dimensions of CX.

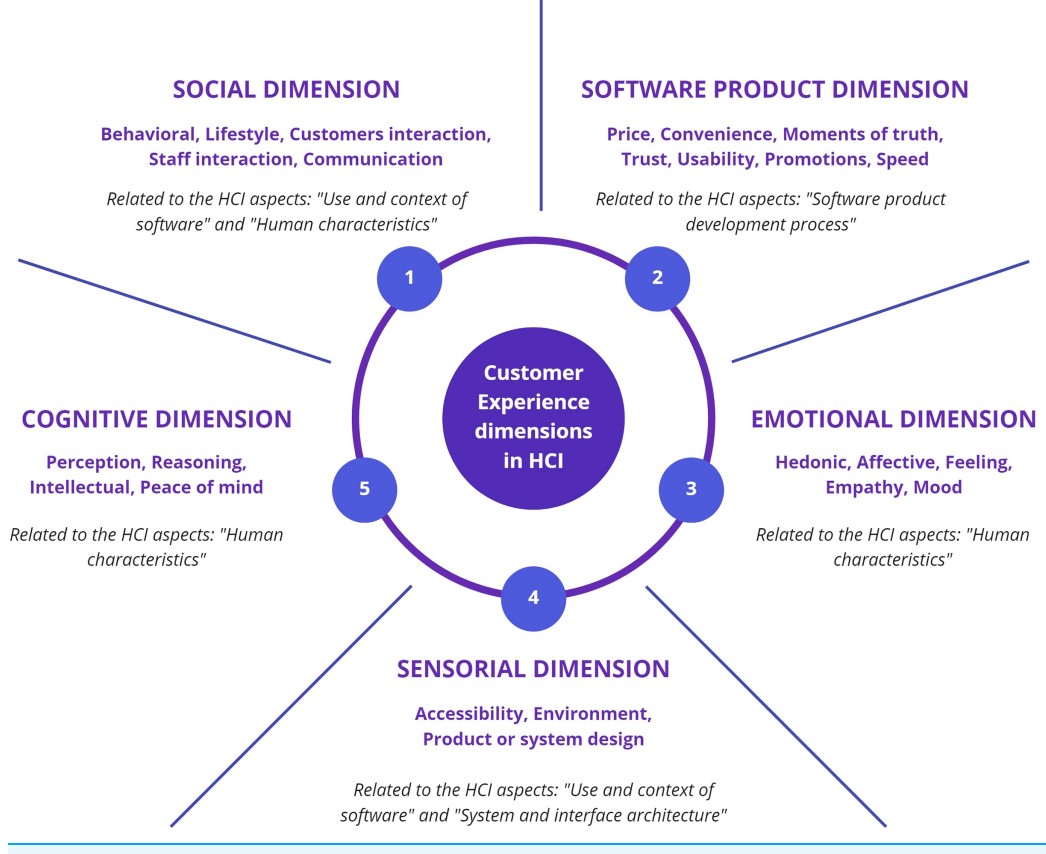

**Figure 12  Customer experience dimensions proposal.**

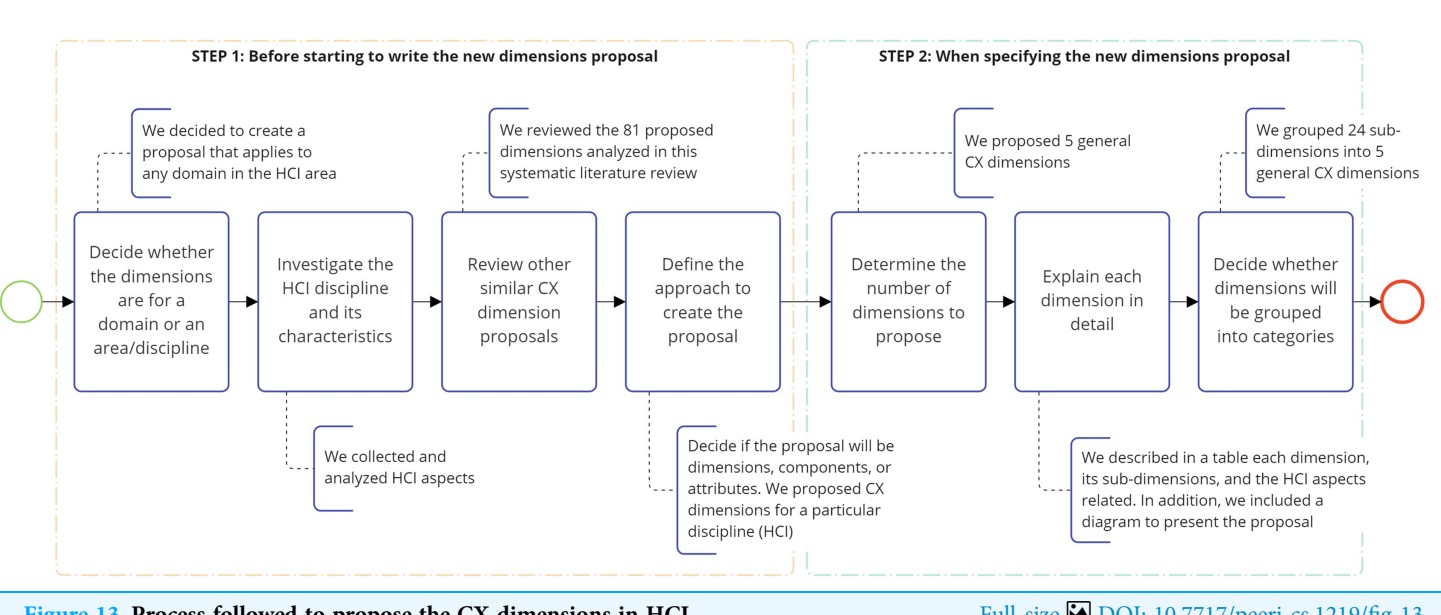

**Figure 13  Process followed to propose the CX dimensions in HCI.**

The five dimensions were chosen since four of them are the most used in dimension proposals covering some HCI aspects (social, cognitive, emotional, sensorial; see Table 14), and the need to incorporate a new dimension covering the HCI aspect "software product development process" (see "About Customer Experience and the Relation with Human-COmputer Interaction (RQ4)"). These dimensions consider the various interactions that a customer may have with a software product over time, in addition to the aspects of the "human being" and context of use that influence their experience. Four of five dimensions address a human component (social, cognitive, sensorial, and emotional), while one dimension considers the characteristics of the software product. All dimensions allow analyzing CX from an HCI perspective. In addition, subdimensions are included for each general dimension, allowing for an accurate understanding of the aspects of CX involved.

This proposal can be applied to domains related to HCI, allowing the design of interactive products and/or systems that generate satisfaction and value for customers. This proposal is an interesting starting point to propose specific dimensions related to HCI. Table 15 presents a detailed explanation of the model proposed, including the dimension definition, its description, its subdimensions; and the related HCI aspects.

This novel CX definition and CX dimensions can be used in domains related to HCI (*i.e.*, domains that involve the interaction of a user—or customer—with a software product) since (1) several articles related to CX (122 studies) were thoroughly reviewed; (2) the studies were analyzed from an HCI point of view; and (3) the HCI dimensions were considered to develop the CX definition and dimensions proposal. This definition and dimensions proposal includes the key concepts about what CX means and can serve as a guide for researchers in the HCI area.

## Implications, novelty, and recommendations for proposing new CX dimensions

The current study suggests important implications for future practice, which can help HCI/CX researchers and practitioners to propose definitions and/or dimensions for the CX in different domains. Regarding CX definitions, this study allows researchers to understand the CX meaning, how it has been conceptualized in several domains, and which elements can be included in a new CX definition based on the most cited definitions. Concerning CX dimensions, this study allows researchers to recognize the common elements in different proposals and what activities they should perform to create new ones. Concerning domains, the most and least studied domains are presented by authors when proposing CX definitions and/or dimensions. This allows researchers to identify and reduce gaps in the CX literature. In addition, to reduce the gap that exists between CX and HCI, a novel definition and dimensions of CX from the point of view of HCI are proposed, considering the components of this discipline. All these implications can be valuable for HCI/CX researchers and practitioners when proposing methods and/or instruments, such as checklists, surveys, questionnaires, scales, or heuristics, to evaluate CX in specific domains, especially in the HCI area.

The findings of this systematic literature review reveal interesting elements that should be considered when proposing new CX dimensions. Eighty-one proposals created by

**Table 15 Customer experience dimensions proposal explained.**

| Dimension | Dimension description | Subdimensions | HCI aspects related (*Hewett et al., 1992*) |
|---|---|---|---|
| Social | Dimension related to the interaction that occurs between people (users, customers, staff, *etc.*). It involves communication between people, their behavior and the interaction that can be generated. | 1. Behavioral: related to the responses and stimuli of the customer to different interactions with a company. <br> 2. Lifestyle: related to the values and beliefs of the customer with a product or system provided by a company. <br> 3. Customer's interaction: related to the customer interaction with other customers. <br> 4. Staff interaction: related to the customer interaction with the company staff. <br> 5. Communication: related to the interaction through different channels and devices such as: websites, customer service, chatbots, *etc.* | "Use and context of software" and "Human characteristics" |
| Cognitive | Dimension related to the customer mental processes (conscious or unconscious). It involves those elements of the product or system that make the customer reason, think and/or interpret things. | 1. Perception: related to the customer awareness and impression about a product, system, or company. <br> 2. Reasoning: related to the logic of customers when interacting with a product, system, or company. <br> 3. Intellectual: related to customers' ability to understand and use products or systems offered by a company. <br> 4. Peace of mind: related to customer evaluation of the different interactions with products or systems offered by a company. | "Human characteristics" |
| Sensorial | Dimension related to the customer's senses. It involves any element of the product or system that generates a stimulus and/or reaction in the customer through the senses. | 1. Accessibility: related to the possibility of interacting with a product or system considering all the customer capacities and/or disabilities. <br> 2. Environment: related to the physical space and its features, such as: location, distribution, aesthetical design, ambiental music, *etc.* <br> 3. Product or service design: related to the product/system "form", such as: color, size, smell, ergonomic, *etc.* | "Use and context of software" and "System and interface architecture" |
| Emotional | Dimension related to the customer emotions and feelings. It involves mood, empathy and any affective and hedonic element that is generated when interacting with a product or system. | 1. Hedonic: related to the intrinsic value and pleasure perceived by customers at different interactions with a company. <br> 2. Feeling: related to the feelings that the customer has when interacting with a product, system, or company, such as: enjoyment, happiness, surprise, sadness, anger, disgust, *etc.* <br> 3. Affective: related to the customer's attitude toward a product, system or company that is influenced by emotions and feelings. <br> 4. Empathy: related to understanding how customers feel about using products or systems offered by a company. <br> 5. Mood: related to the temporary state of mind or feelings that the customer has when interacting with a product, system, or company. | "Human characteristics" |

(Continued)

| Table 15 (continued) | | | |
| --- | --- | --- | --- |
| Dimension | Dimension description | Subdimensions | HCI aspects related (*Hewett et al., 1992*) |
| Software product | Dimension related to the features of the product and/or system offered by a brand or company. It involves all the aspects that can influence the customer's experience and their perception of what is offered. | 1. Price: related to how customers perceive the value of the products or systems offered by a company. <br> 2. Convenience: related to the ease of customers to obtain products or systems offered by a company. <br> 3. Moments-of-truth: related to the attitude and actions of the company to customers when complications arise. <br> 4. Trust: related to the confidence that customers have in the products and systems offered by a company. <br> 5. Usability: related to the ease of use of products or systems offered by a company. <br> 6. Promotions: related to the benefits given by the company to attract or maintain customers. <br> 7. Speed: related to both the speed of the product or system purchased and the waiting times. | "Software product development process" |

different authors in several domains were detected. As discussed earlier in this section, several of these dimensions have been based on previous proposals, and some of them have certain deficiencies that may make it difficult to understand, design, or evaluate CX. When creating a new dimension proposal for the CX, performing at least the following activities grouped into three steps or phases is suggested:

1. Step 1: Before starting to write the new dimensions proposal

1.1. Decide if the proposal is for: (1) a domain (main or specific domain); or (2) an area or discipline. If the proposal is for a domain, identify the domain characteristics. Understanding the domain and its characteristics allows for the creation of dimensions that are related to the domain. If the researcher decides to create a proposal that applies to any domain, then he or she should determine in which area or discipline it will be framed (engineering, psychology, human-computer interaction, among others). In this case, investigating the discipline and its characteristics is recommended.

1.2. If the proposal is for a domain, then identifying the most relevant touchpoints for that domain is advised. This will allow for knowing the moments in which the customer interacts with a product, system, or service of a brand or company, mapping the customer journey, and defining dimensions for each moment of interaction.

1.3. Review other similar CX dimension proposals (either general or specific for the domain or discipline studied). A critical analysis of other proposals is suggested (for example, their strengths, weaknesses, benefits, and what can be improved, incorporated, or adapted; that is, what can be taken as a basis to create the new proposal).

1.4. Define the approach to use to propose the dimensions:

1.4.1. Does the researcher want to propose dimensions for the entire customer journey, or for one or more specific touchpoints? This will help to define how to raise the dimension proposal.

1.4.2. Does the researcher want to consider aspects, properties, or qualities that will characterize the CX? If so, then what will be proposed will be CX dimensions or attributes. This proposal will be more specific, it will only apply to the particular domain and will be oriented to design and/or evaluate the CX for those aspects.

1.4.3. Does the researcher want to propose elements that must be "put together" in a certain order and way to create an experience? If so, then what will be proposed will be CX components, which will correspond to a more general proposal that encompasses different "experiences".

2. Step 2: When specifying the new dimensions proposal

2.1. Establishing between four and six dimensions is recommended. As we indicated in "About Customer Experience Dimensions and Domains (RQ2 and RQ3)", proposals with few dimensions (less than four) may be too general, ignoring important CX elements, and proposals with many dimensions (more than six) may be too specific, can only be used for the specific domain, or even lead to confusion.

2.2. Explain each dimension in detail for better understanding (especially for future studies or researchers who want to use the proposal). In addition, it could be useful to include a diagram that explains the relationship between the dimensions (if applicable).

2.3. Determine whether or not to include examples to better understand each proposed dimension.

2.4. Determine if the dimensions will be grouped into categories (which ones and why).

3. Step 3: After raising the new dimensions proposal

3.1. Validate the new proposal through experiments (with customers and experts), case studies, statistical analysis, and/or expert judgment. It is important to determine if the proposal correctly considers the characteristics of the domain or discipline and if the relationship between the dimensions is adequate.

3.2. Refine the proposal based on the feedback obtained in the validations. The number of validations and/or refinements will depend on the results obtained. At least two validations are suggested. Some questions to ask yourself to plan the validations can be:

3.2.1. Are all the characteristics of the domain or discipline considered in the proposal? Are all of those features addressed or covered?

3.2.2. Do the dimensions cover the most important elements of the entire customer journey in the domain (touchpoints)?

3.2.3. Are at least all UX dimensions or factors included? This, considering that CX is an extension of UX (*Rusu et al., 2020*; *Lewis, 2014*).

The activities proposed above are based on the steps that other authors have followed to propose dimensions (based on the articles analyzed in this systematic review). Our activities condense and consolidate the activities that other authors have carried out to create their proposals.

We used these same activities to propose the CX dimensions presented in "A Novel Customer Experience Definition and Dimensions for Human-Computer Interaction" (except for stage 3). Although the proposed activities have not been applied in other domains, it is believed that they facilitate the process of developing new dimensions. The above activities are useful to create a valuable and effective proposal of CX dimensions that consider all the main features and make it possible to apply them in future studies, especially in the HCI area.

### Limitations

The results obtained in this review are subject to certain limitations. For instance, the literature review was narrowed by the defined search terms, the database considered, and the time range of the studies published. In addition, some studies could not be reviewed due to the unavailability of their sources. Another limitation is that the review was performed by two authors, who reviewed the articles at the same time in different databases. After tabulating all the information, some discrepancies were detected in the data that had to be corrected, such as the year of publication, the application domain, or the term used to name the domains (in some cases, the "main domain" name was defined, since different authors used synonyms). However, a thorough review of the tabulated data was conducted to ensure that there were no inconsistencies or incorrect information.

Despite the limitations, this review provided a wide view of the CX definitions proposed by different authors in various domains, as well as the different CX dimension proposals. Therefore, the review represents the current state of the literature. Additionally, the most appropriate databases were considered, and the most pertinent terms related to the review were used. The systematic literature review covered studies published over a period of 10 years (2010 to 2020), even including studies before this period, as they are highly cited and considered extremely relevant in the CX area.

## CONCLUSIONS

The main goal of the current systematic review was to identify studies proposing CX definitions and/or dimensions to contextualize CX in different domains and analyze the results from an HCI perspective. Four research questions guided this review: (1) what the CX is; (2) what dimensions define the CX; (3) in which domains the CX concept is used; and (4) what the relation between CX and HCI is. After reviewing 122 studies, 71 CX definitions and 81 CX dimension proposals were identified. In addition, 30 main domains and 79 specific domains in the selected studies were recognized.

The most used CX definition in the literature corresponds to the proposal by *Meyer & Schwager (2007)*. This is mainly because they provided two novel aspects within the proposal that had not been mentioned: (1) the internal and subjective response of customers; and (2) the direct and indirect contacts with companies. Unexpectedly, despite

the year of publication of the study, the most used dimension proposal in the CX literature corresponds to *Schmitt (1999)*. The author introduces the dimensions (named by the author "modules") of sense, feel, think, act, and relate. Several studies cite this proposal since it was apparently the first to propose CX dimensions. In addition, it is widely used as a basis in multiple studies that propose new CX dimensions for specific domains.

Some domains were studied widely, such as Retail, and others narrowly studied, such as the Restaurant or Education domains. Specifically, concerning the studies that propose CX definitions, 24 main domains, and 45 specific domains were identified, where the main domains with the most proposed CX definitions are Retail, Telecommunication, and Management. For the studies that propose CX dimensions, 31 main domains, and 57 specific domains were found, where the domains with the most proposed CX dimension proposals are: Retail, Bank, and Hotel.

HCI was studied from a CX perspective since customers interact with companies through several products or systems that may be related to software systems. Regarding the relation between HCI and CX, 14 CX definitions that highlight aspects of HCI in 10 different domains were identified, such as retail, market, and e-commerce. On the other hand, 24 proposals containing CX dimensions that could be related to HCI in 12 domains were recognized, such as Telecommunication, Digital technology, and Digital platform. In addition, 18 domains were found that contained CX definitions and/or dimensions that could be related to HCI, where the domains that present more proposals of CX definitions and dimensions related to HCI are Retail and E-Commerce.

Furthermore, this systematic literature review contributes to existing knowledge of CX and HCI by providing an extensive overview of what CX means and what dimensions have been proposed to describe it. Moreover, it provides a detailed review of the different domains in which CX is studied and analyzes how studies have been increasing over time. Finally, this review provides a novel CX definition and dimensions from an HCI perspective, with recommendations for proposing new CX dimensions in domains related to HCI (*i.e.*, domains that involve the interaction of a user—or customer—with a software product). The definition and dimensions of this study include the most used and relevant concepts, adding new elements that are necessary to understand the multidimensional and holistic vision of CX. The recommendations include performing different essential activities grouped into three steps: (1) before starting to write the new dimensions proposal; (2) when specifying the new dimensions proposal; and (3) after raising the new dimensions proposal.

Concerning future research, the proposal of new CX dimensions focused on HCI for a specific domain can be developed to reduce the gap between both areas. In addition, a systematic literature review can be performed to investigate what methods or instruments are used to evaluate CX in the HCI field, what dimensions are evaluated, and how the applied methods vary depending on the context studied. Additionally, it may be interesting to develop instruments that allow the evaluation of CX from an HCI point of view. Since there are no specific studies that establish CX dimensions in interactive software products that consider users as customers, the results of this systematic review show that there are research opportunities in the field of HCI and CX. Furthermore, the results obtained in this

systematic review can be complemented by consulting other databases or focusing the review on a particular domain relevant to HCI but less studied, such as virtual communities or smart services.

### Funding
Daniela Quiñones is supported by Grant ANID, Chile, FONDECYT INICIACIÓN, Project No. 11190759. Luis Rojas has been granted the "INFPUCV" Graduate Scholarship. Luis Rojas is supported by Grant ANID BECAS/DOCTORADO NACIONAL, Chile, No. 21211272. The funders had no role in study design, data collection and analysis, decision to publish, or preparation of the manuscript.

### Grant Disclosures
The following grant information was disclosed by the authors:
ANID, Chile, FONDECYT INICIACIÓN: 11190759.
"INFPUCV" Graduate Scholarship.
 ANID BECAS/DOCTORADO NACIONAL, Chile: 21211272.

### Competing Interests
The authors declare that they have no competing interests.

### Author Contributions
- Daniela Quiñones conceived and designed the experiments, performed the experiments, analyzed the data, prepared figures and/or tables, authored or reviewed drafts of the article, and approved the final draft.
- Luis Rojas performed the experiments, analyzed the data, prepared figures and/or tables, authored or reviewed drafts of the article, and approved the final draft.

### Data Availability
The customer experience definitions and dimensions are available in the Supplemental Files.

### Supplemental Information
Supplemental information for this article can be found online at http://dx.doi.org/10.7717/peerj-cs.1219#supplemental-information.

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
