# Peer review of "Understanding the customer experience in human-computer interaction: a systematic literature review"

_PeerJ Computer Science, doi:10.7717/peerj-cs.1219_

## Round 0.1 · original submission · Minor Revisions

Please see the reviewers' comments. In particular, reviewer 1 provided valid comments. In addition to reviewers' comments. I believe the section 3 research questions information should be included in the Introduction section following the study's main aim. There is no need for a separate section including the table-2 of research questions.

Reviewer 1 ·

Basic reporting

I think this is a pretty good paper, informative, with substantial findings, and reflecting a substantial amount of work. The paper is well-organized and formatted and generally well-written. It presents a clear argument for customer experience to be considered within the various domains of HCI. The authors present sufficient background and literature, while utilizing this body of work to support their conclusions. Some articles referenced in this paper are rather old, some being up to 20 years old, but this is okay, considering the context of the discussion presented in the paper. My recommendations are to:
(a) Further proofread the paper and improve English a bit as some errors still exist. For example, in the abstract the sentence "the guidelines proposed by Kitchenham were used, complementing it with a snowballing approach" is incorrect (what's the "it" here when the subject is clearly plural, "guidelines"?). Also, the header of 6.1 "What is the customer experience/customer experience definitions?" needs to be revised because "is" doesn't apply correctly to the second part, "definitions". In some cases the tenses need to be used more consistently/appropriately. For example, in the paragraph that starts Section 3, all work described has been completed -- it's now in the past. So, present tense needs to be changed to past tense in a couple of places, specifically "they allow" to "they allowed" and "we do not leave" to "we did not leave". As I said, please check again the entire paper to identify and fix potential other similar occurrences.
(b) The mapping between the four research questions in Table 2 could be mapped more straightforwardly into the header titles for subsections in Section 6. Titles for subsections are, in my view:
6.1 OK, but has an English error (as indicated above).
6.2 Needs improvement. RQ2 talks about "attributes and dimensions" but in the header of 6.2 and the related text "attributes" have vanished.
6.3 Needs improvement. Section header 6.3 can be mapped better to the research question, for example by changing it to "Domains in which the customer experience concept is used."
6.4 OK .

Experimental design

As this paper is a literature review, it fits the Aims and Scope of the journal. The survey methodology is organized and presented into a readable and understandable way.

Validity of the findings

The impact and novelty of CX vis-à-vis HCI are well addressed in this paper and support the idea that the user could be considered a customer. The conclusions are well stated and link to the authors' original research goals.

Additional comments

Could there be any limitations from a design perspective of viewing users as customers?

Reviewer 2 ·

Basic reporting

English is clearly written

References seem up to date but authors should check in revision

Professional structure is there

Experimental design

Novelty can be explained better

Validity of the findings

Authors seem to present this clearly

Additional comments

In this paper, the authors present an extensive review of 122 studies related to CX defnitions and
dimensions that have been proposed in different domains, including an analysis from an HCI perspective.

Revisions as follows:

a) Readability should be improved

b) Figures quality should all be HD

c) Can tables etc be presented inline to see placement

d) Please check PeerJ author guidelines for Refs

e) THere are far too many Tables, it is hard to follow

---

## Round 0.2 · Minor Revisions

The reviewer is happy with the last revision, but there are a couple of comments. Please see the attached.

Reviewer 1 ·

Basic reporting

The authors have addressed very thoroughly all my comments and recommendations from my initial review. As I said then, the article is on an interesting and meaningful topic, reflects a substantial amount of work, and provides good contributions. After the revision I think the article has been further enhanced and presents in a well-written and well-structured way a significant research effort. While all tables and figures are meaningful, the findings and results depicted in Figures 11, 12, 13 are particularly valuable in my view. Other key contributions of the article are its Discussion section, which includes a new and useful definition of customer experience (CX) and five proposed dimensions for describing CX from an HCI perspective. My observations now are the following:

1. In Figure 12 use consistently "related to the HCI aspects" (in some places only "aspect" is used);
2. The proposed definition of CX on lines 759-770 is good, but too long for effective referencing by peers and other readers. I really like the related journey metaphor (with touchpoints, which are moments of interactions) and would capture this in the early part of the definition. A suggestion is to highlight in italics or bold the first sentence of the definition ("The customer experience corresponds ...") , add immediately after (I suggest also in italics or bold) something along the lines "It emerges from a customer journey that consists of three distinct yet interconnected stages: pre-interaction, interaction, and post-interaction." Then continue with the current sentences on lines 761-770 (in some slightly revised form), not in italics or bold, and thus meant as a supporting or secondary part of the definition. Again, these are just my suggestions, but please feel free to keep the definition as it is now, if you think that's the best way to formulate it. Note that on line 767 it should be "conform to the customer journey" ("to" is missing).
3. In general, in English texts one-digit numbers are spelled out with letters, like "three," "four," or "seven" but I understand and actually like using numerical figures such as 3, 4, or 7 to highlight quantitative aspects of some entity or topic. However, I suggest not to start a sentence with just a number, thus I'd change on line 789 the current wording to "Four of the 5 dimensions address ..."

Experimental design

As this paper is a literature review, it fits the Aims and Scope of the journal. The survey methodology is well-organized and presented in a readable and understandable way.

Validity of the findings

The impact and novelty of CX vis-à-vis HCI are well addressed in this paper and support the idea that the user could be considered a customer. The conclusions are well stated and "link back" well to the authors' original research goals.

Additional comments

Good, solid work, with interesting contributions. Please check my few final recommendations in the "Basic reporting" segment of my review.

---

## Round 0.3 · accepted · Accept

The authors have addressed all of the reviewers' comments. I believe the manuscript is ready for publication.